# Coastal extreme sea levels in the Caribbean Sea induced by tropical cyclones

Ariadna Martín[1], Angel Amores[1,2], Alejandro Orfila[1], Tim Toomey[1], and Marta Marcos[1,2]

[1]Mediterranean Institute for Advanced Studies (UIB-CSIC), Spain
[2]Department of Physics, University of Balearic Islands, Spain

**Correspondence:** Marta Marcos (marta.marcos@uib.es)

**Abstract.** Every year the Caribbean Sea faces the passage of several tropical cyclones that generate coastal extreme sea levels with potential strong and hazardous impacts. In this work we simulate the storm surges and wind-waves induced by a set of 1000 tropical cyclones over the Caribbean Sea that are representative of the present-day climate. These events have been extracted from a global database of synthetic tropical cyclones spanning a 10,000-year period. The atmospheric forcing fields, associated to the set of tropical cyclones, are used to feed a coupled hydrodynamic-wave model with high resolution ($\sim$2 km) along the continental and islands coasts. Given the large number of events modeled, our results allow detailed statistical analyses of the magnitude and mechanisms of coastal extreme sea levels as well as the identification of most exposed areas to both storm surges and extreme wind-waves.

## 1  Introduction

Tropical cyclones (TCs) are among the most hazardous natural disasters, significantly affecting population safety, economies and ecosystems in coastal areas. The Eastern coast of Central and North America are among the most affected regions to these events worldwide (Needham et al., 2015). Here we focus on the coastal regions of the Caribbean Sea, an area far less investigated when compared to the Gulf of Mexico, in spite of its exposure to TCs being similar. With a relatively small landmass and total population ($\sim$ 44 million people based on the latest United Nations estimates (woldometers, 2022)), the Caribbean has experienced, in the past two decades, 163 storm events (hurricanes and storm surges) affecting a total of 25.8 million people and resulting in more than 5000 deaths (Centre for Research on the Epidemiology of Disasters, 2020). Moreover, the Caribbean Sea contains more than 700 islands. The characteristics of these tropical islands (i.e their size, coastal population density, morphology, elevation and coastal defences) make them particularly vulnerable territories to the impact of low-frequency and high-intensity events, such as TCs (Pillet et al., 2019b; Duvat et al., 2017; Giuliani and Peduzzi, 2011). Even for some of these small island nations, the damages caused by TCs can exceed the size of their economies. This is the case of Dominica, that in 2017 was affected by Hurricane Maria causing a damage cost estimated of 226% of its 2016 gross domestic product (GDP) (GFDRR, 2017). Another example is Grenada, affected by Hurricane Ivan in 2004 that caused an economic damage exceeding 200% of their GDP (ECLAC, 2004). As the socio-economy of small islands depends on the preservation of the coastal zone, it is crucial to investigate and support hurricane risk mitigation efforts in these areas.

TCs are a relatively rare phenomenon with around 90 ($\pm$ 10) formations per year globally. In addition, historical records of TCs are in general scarce, since reliable TC dataset are only available from 1980 onwards (Bloemendaal et al., 2020). In consequence, there have been earlier studies that are limited to particular small areas, islands or bays, where observations are available during the passage of a TC. This is the case for example of Batabanó, Samaná Bay, Guadeloupe and Dominica (Arenal et al., 1998; Krien et al., 2015). One way to overcome the historical lack of observations of TCs is the generation of synthetic

hurricane databases, built consistently with the observed TCs characteristics that allow more robust statistical analyses of TCs frequency and intensities and that can be used to evaluate the associated risks (Appendini et al., 2017; Lin et al., 2014). This study will take advantage of one of these synthetic hurricane databases by Bloemendaal et al. (2020) that is used here to provide an unified Caribbean-wide view of the marine hazards generated by TCs. This dataset contains hurricanes and tropical storms; therefore we are going to refer to all the dataset as TCs. The amount and geographical consistency of synthetic data permit to

perform a spatially coherent study along all Caribbean coastlines, instead of investigating single events. In particular, we focus on the ocean hazards generated by TCs in terms of wind-waves and storm surges, using a set of 1000 TCs extracted from the synthetic dataset combined with a fully coupled hydrodynamic-wave model. This is the first time, to our knowledge, that both mechanisms responsible for coastal hazards have been investigated together and at basin scale in this region in a consistent manner. We analyse in detail the outputs of the numerical simulations to quantify the role of the different forcing factors to

coastal extreme sea levels from TCs, namely, the atmospheric pressure, the wind and the wind-waves.

This paper is structured as follows: in section 2 we describe the different datasets used together with the hydrodynamic model. Section 3 include the results of wind-waves and sea surface elevation along the Caribbean coastlines. Summary and conclusions are given in Section 4.

## 2   Data and Methods

### 2.1   Data

The tracks of synthetic TCs were extracted from STORM (Synthetic Tropical cyclOne geneRation Model) dataset (Bloemendaal et al., 2020). It consists of 10000 years of TCs over the globe obtained using a fully statistical model approach fed with observed TC tracks and, thus, being representative of present-day climate conditions. This dataset provides, for each TC, its track, minimum pressure, maximum wind speed, and the radius of maximum wind speed (Rmax) with a 3-hourly time step

during its lifetime. Within the 10000-year synthetic TC available in the STORM dataset, 25494 of them affected the Caribbean Sea ($\sim 2.5\,\mathrm{TC/year}$). This rate is in agreement with the 2.7 TC per year observed from 1944 to 1970 (Goldenberg et al., 2001) and the 2.47 TC per year published in Chenoweth (2006) computed from 155 years of records. Among the entire set of TC affecting the Caribbean, we aim at selecting a subsample in such a way that it is statistically consistent with the original sample in terms of the TC intensity and its spatial distribution (Fig. 1) and, at the same time, small enough as to perform basin-wide

numerical simulations. There are several methods to select a subsample that is consistent with a larger data set (e.g., K-means (Enríquez et al., 2020), Maximum Dissimilarity Algorithm, (Camus et al., 2011) ). Among the different options, we follow Toomey et al. (2022), to choose the size of the subsample. This consists of comparingthe probability distribution functions

(PDF) of the original dataset (corresponding to the 25494 TCs) to the selected samples (with sizes ranging between 10 to 10000 events) on the basis of two statistical tests, namely the root-mean-square-deviation (RMSD, Fig. 1a) and the correlation (Fig. 1b). This process was performed 1000 times for each size of the subsamples (selecting randomly the events each time) and for both the maximum wind speed and the spatial distribution of the TCs tracks PDFs. In this case the PDF is built using the number of TCs (counting each time step) passing through each pixel of the grid. The spatial domain has been divided into 2 degrees bins, represented in Fig. 1, while for the maximum wind speed distribution the bins size was fixed at 2m/s. We seek to determine a size for the subsample that is a good representation of the original distribution and also small enough as to perform the numerical simulations described in section 2.3 within a reasonable computational time. Our results (Fig. 1a and c) indicate that a subsample of 1000 TCs is statistically consistent to the original distribution, both in intensity and spatial distribution in this area. The maps in Fig. 1 represent this behaviour, showing the percentage of TCs per pixel for the total sample (panel e) and the selected sub-sample (1000 TCs, panel f).

In addition to the synthetic TCs described above, five real events were studied (see Table 1). These have been selected for model validation because observations for buoys and/or tide gauges were available. To recreate the atmospheric fields of these events, the maximum wind speed and minimum pressure along the track were extracted from IBTrACS (Knapp et al., 2010, 2018), with a temporal resolution of 3 hours, so the output is directly comparable with the synthetic data. The radius of maximum wind speed ($Rmax$) is also necessary to build the associated 2D fields (see section 2.2). This is available in IBTrACS only after year 2000, so all five selected events belong to the last 20 years of data.

Coastal sea-level observations from tide gauges and wind-wave measurements from in-situ buoys have been used to validate the numerical simulation of the already mentioned five real hurricanes. Sea level data were obtained from GESLA dataset version 2 (https://gesla.org/, Woodworth et al. (2016)) which contains a total of 46 sea level records in the Caribbean Sea. Unfortunately, given their location they contain very few footprints of TCs, further reduced due to the limited period after year 2000. We have considered only peaks of sea level exceeding 0.4 m. Only three TCs in two different tide gauges fulfill this criterion for this period of time. To compare with the sea surface elevation, SSE, of the model that only accounts for the contribution of the TC, we compute the non-tidal residual using UTide function for the complete record. The wind-wave measurements were retrieved from the National Data Buoy Center (1971) (NOAA, https://www.ncei.noaa.gov/archive/accession/NDBC-CMANWx, last access: 2 September 2020). A total of 34 buoys were available inside the area of study 27 of which are found inside the Caribbean Sea, and 7 in the nearby Atlantic Ocean. In contrast to tide gauge data, more wave observations were available from buoys. This adds two more TCs, resulting in a total of five TCs and 15 buoys used for the validation of wind-waves. All instruments selected are shown in Fig. 2.

## 2.2 Generation of 2D fields from synthetic tracks

The 1000 synthetic TCs tracks extracted from STORM dataset were used to built the 2D atmospheric fields required to force the hydrodynamic model. Atmospheric pressure and surface wind fields for each TC were generated onto a 5-km regular grid using the same temporal resolution of the synthetic tracks (3 hours). The wind field accounts for a translational component that drives the TC forward and a cyclonic component. The former was calculated using the track, while the radial component was

| TC | Buoys | Tide Gauge |
|---|---|---|
| ♦ Wilma (2005) | 42057,42056 | - |
| ▲ Tomas (2010) | 42059,42060 | - |
| ● Ernesto (2012) | 41040,41043,42057,42056,42058,42059,42060 | Puerto Moreles |
| ■ Omar (2008) | 42058,42059 | Christiansted |
| ★ Ida (2009) | 42057,42056 | Puerto Moreles |

**Table 1.** List of the TCs used for the validation along with their respective buoys and tide gauges used in the validation.

calculated based on the empirical Holland wind profile (Holland, 1980) with the latest revision of the formulation (Holland et al., 2010). This formulation has been extensively used for reconstructing hurricane wind fields (Sitkowski et al., 2011; Boose, 2004). Translational and circular wind velocity components were added, after setting the translational component to zero at

distances greater than 300 km from the hurricane eye (Holland, 1980; Holland et al., 2010). The final step for the wind fields consists of reducing the velocity by 20 % over land areas (Willoughby and Black, 1996).

### 2.3 Hydrodynamicl model

We simulated storm surges and wind-waves caused by the 2D fields derived from the synthetic and the five real TCs over the Caribbean Sea using the latest version (5.8) of SCHISM model (Semi-implicit Cross-scale Hydroscience Integrated System

Model, Zhang et al. (2016)), a state-of-art cross-scale hydrodynamic model, which is a derivative product built from the original SELFE (Zhang and Baptista, 2008). Here we have used the hydrodynamic core in its depth-average (2DH) barotropic mode fully coupled with the spectral wave model WWM-III (Roland et al., 2012).

The bathymetry used for the hydrodynamic model and the spectral wave model was GEBCO 2020 (available at: https://www.gebco.net/data_and_products/gridded_bathymetry_data, (GEBCO, 2020)) on a 15 arc-second geographic latitude and

longitude grid, which in the Caribbean Sea region corresponds to ∼ 450 m. Both models were implemented using the same unstructured grid (generated using the Triangle algorithm (Shewchuk, 1996)) that covers the whole Caribbean Sea and part of the Atlantic Ocean with a total of 135759 nodes (Fig. 2). The spatial resolution varies as a function of the depth, within a range of around 40 km in open ocean down to ∼ 2 km along the coastlines. The coastline used was downloaded from https://osmdata.openstreetmap.de/data/coastlines.html which, on average, has a spatial resolution of approximately 40 m (OpenStreetMap

contributors, 2017). The resolution of the coastline was reduced to 2 km, except in areas with abrupt changes in the coastline.

The computational domain (Fig. 2) was chosen to include the prevailing TC incoming directions. Although TCs affecting the Caribbean Sea can develop further inside the Atlantic Ocean, from the coast of West Africa, the selected domain is large enough to allow a correct generation and propagation of the wind-waves originated by TCs affecting the Antilles (i.e. it accommodates enough space for the wind fetch to act). There are two open boundaries in the model, both with free elevation and velocity

forced to 0, corresponding to the Atlantic ocean boundary and the region between Cuba and Mexico (black lines Fig. 2).

The spectral domain for the wind wave model has been fixed to 24 bins for both direction (i.e., 15° bins) and frequency (ranging between 0.04 to 0.6 Hz). For the bottom friction the Manning's drag coefficient was fixed to 0.02 and the wind stress was calculated using Pond & Pickard formulation (Pond and Pickard, 1983), used for the hydrodynamic model, while for the fully coupled run the wind surface stress is calculated directly using the forcing fields which has proved to be superior to the former when waves are available (Bertin et al., 2015). The computational time step was set to 10 minutes for the hydrodynamical model and 30 minutes for the wave model. The selected variables (significant wave height ($H_s$), sea surface elevation, air pressure, wind speed, wave peak period ($T_p$), and wave peak direction ($D_p$)) were saved every hour. The model configuration was defined after model validation using the 5 real events, with a reasonable computational cost (see Sec. 3.2).

To analyse the role of the different contributions of the atmospheric forcing fields, i.e. atmospheric pressure, wind and the wind-waves, four different numerical simulations where performed for each TC: (1) a fully coupled run between the hydrodynamic model and the wind-wave module that takes into account all contributions as well as their coupling; (2) a run using only the hydrodynamic model forced with atmospheric pressure and wind; here the contribution of the wind-waves is excluded; (3) a hydrodynamic model run forced using only the atmospheric pressure, the winds were fixed to 0 m/s; and (4) a hydrodynamic model run forced only by the wind, setting the atmospheric pressure constant to 101325 Pa. Finally, the contribution of the wind-waves was extracted from the differences between runs number 1 and 2 which accounts for all the couplings, while the effects of the atmospheric pressure and wind could directly be quantified from the simulations 3 and 4, respectively. The method for obtaining the pure contribution of each factor, as well as the synergistic effect due to the mutual interactions among two or more of these factors should be carried out using $2^n$ simulations, where n is the number of forcings to be separated. Thus, the wind and pressure contributions were correctly separated, while the wind-waves contribution implicitly includes all the couplings with the other forcing interactions (wind-waves - wind, wind-waves - pressure and the wind-waves - pressure - wind term), of which we assume are small compared to the main contribution of wind-waves (Amores et al., 2020).

The computational time of each TC was $\sim$ 3 hours for the fully coupled runs using only one CPU. In our case, we used 20 processors which allowed us to reduce the computational time to approximately 10 days by running simultaneously 20 TCs. For the hydrodynamic runs in which the wind-wave model is not used, the total computational time was about two days per configuration (including the 1000 TC). This leads to a total computational time of approximately 2 weeks.

## 2.4 Computation of return levels

Return levels for extreme coastal waves and sea surface elevation (SSE) at every coastal grid point have been calculated using a Generalized Pareto Distribution (GPD) fitted to all values over a chosen threshold. Using POT (Peaks Over Threshold) analysis together with GPD, increases the number of measurements included in the analysis, and correspondingly reduces the statistical uncertainty of quantile variances (Brabson and Palutikof, 2000). Studying return periods for events such as TCs adds a difficulty in selecting the threshold, since by definition they are all extremes. However, each event affects different areas of the Caribbean Sea, so that is not appropriate to set a common threshold for the entire area of study; rather, the threshold has been chosen for every grid point. For this purpose, the data at each grid point have been adjusted to a GPD successively, each time removing one value from the adjustment. The threshold is considered as the point where the GPD stabilises and all the

return levels remain constant regardless the value of threshold. We consider this point as the threshold of the extremes of each grid point.

## 3  Results

### 3.1  Characteristics of TC affecting the Caribbean Sea.

The characteristics of the TCs affecting the Caribbean coastlines are represented in Fig. 3 in terms of their intensity and
frequency. The top panels show the median (a) and $95_{th}$ percentile (b) of the maximum wind speed of TCs reaching the coastlines (note the different colour scales). The number of TCs affecting the coasts (and for which the speeds are represented in the upper panels) is mapped in panel c, showing a well defined geographic pattern: while the eastern part of the basin receives more than 3 TCs per decade, the western coasts are affected on average by less than 1 per decade. The lower impacts in this region can be related to the decrease of the Coriolis force towards the equator and the consequent low probability, less than 1%
160  per year (Torres and Tsimplis, 2014), of TCs turning southwards to these coasts. The southern coast of the Caribbean (north of South America) barely sees any TC because of their prevailing travelling direction towards the west-northwest which suggests that the energy is dissipated before reaching those coasts. The areas affected by a larger number of TCs include the West Indies, the islands of La Española (Dominican Republic and Haiti) and Puerto Rico, which are heavily exposed to Atlantic TCs. Given the clear spatial differences, it is important to distinguish two different families of TCs, that can be classified depending on their
region of origin, either in the Atlantic Ocean or inside the Caribbean Sea. Each one of these families has a distinct footprint along the Caribbean coastlines, with the eastern part of the Caribbean Sea being affected only by TCs generated in the Atlantic Ocean and the western coasts being hit by both families although primarily by those originated within the same basin. This pattern is displayed in Fig. 3e and f where the median intensities of the two families of TCs affecting each coastal point are represented. The TCs from Atlantic Ocean (panel f) are the more intense reaching maximum speeds exceeding 200 km/h,
also more frequent 66.1 %, while these values are around 125 km/h, and 33.9% for TC generated within the Caribbean (note the different colour scales in the maps). Interestingly, the most intense TCs hit the western part of the Caribbean coastlines; the reason is that these TCs, that are strong in origin, are further intensified during their trip through the warm Caribbean Sea. Regarding the areas exposed to both families of TCs, the western part of Cuba is the area hit by the most intense TCs. Although the TCs affecting the lesser Antilles are not the most intense ones, the greater number of them makes this area one of the most
exposed to their impacts.

These results have been obtained using the TCs from the selected subset. When the same maps are produced with the complete TC dataset, the patterns are essentially identical with the only exception in the number of Caribbean TCs and their intensities affecting the Antilles (see Fig. S1c). The difference comes from a few TCs in the complete dataset that are generated into the Caribbean Sea and move eastwards, instead of westwards predominant direction. This is an extremely rare
phenomenon, accounting for less than 20 TCs from the 25494 available in the dataset, that represents less than 0.001% of the total number of TCs generated in the Caribbean Sea. Thus, it is not surprising that they are not represented in the subset. Landfall patterns are a very important factor in the calculation of TC risk. Figs. 3 and S.1 show a clear agreement between

the subset and the complete dataset from STORM. However, there are some discrepancies in the number of landfalls per year between STORM and IBTrAcs, due to a substantial difference in the year-to-year landfall counts of IBTrACS dataset. Yet, on average, landfall counts of the two datasets are within one standard deviation of each other (Bloemendaal et al., 2020).

## 3.2  Model Validation.

The performance of the methodology, including the construction of synthetic atmospheric fields and the ocean model, has been tested against all available in-situ observations of $H_s$, $T_p$ and SSE. To do so, all TCs within IBTrACS for which maximum wind speed, minimum pressure along the track and radius of the maximum wind speed are available have been used. This limits the number of events, since radius is only available after year 2000. The TC that fulfil these criteria and for which there exist in-situ observations are listed in Table 1. Results are plotted in Fig. 4 in terms of maximum modelled and observed values. For the model, the closest grid point to the buoy or the tide gauge is chosen. Note that the number of observations for $H_s$ is significantly larger than those for SSE. This is because there are few TCs that have passed close enough to a tide gauge since year 2000 (Torres and Tsimplis, 2014). Our criterion is that at least a sea level peak of 0.4 m must be recorded to be linked to a TC.

In general, there is a good agreement between observed and modelled maxima, especially for the wave data, panel a) and b). We therefore conclude that the methodology is accurate enough as to represent the response of the ocean to incoming TC in the Caribbean Sea. Shortcomings of the results and the model setup will be discussed later in the last section.

## 3.3  Coastal wind-waves induced by TCs.

The metric chosen to characterise the wave hazards induced by TCs along the coastlines has been defined as the $99^{th}$ percentile of the set of maximum $H_s$ for each TC at every coastal grid point (i.e., a measure of the tenth most intense TCs from the 1000 simulated for each particular location). The results are mapped in Fig. 5, which includes an inset with a zoom in the Lesser Antilles. The highest values, exceeding 10 m and reaching almost 20 m in some areas, are found in the West Indies, including the islands of La Española (Dominican Republic and Haiti) and Puerto Rico, and in the Caribbean coast of Mexico. It is worth noting that the highest waves hitting the Lesser Antilles (Fig. 5b) are found in their eastern coasts, as expected as they are mainly hit by TCs of Atlantic origin travelling westwards. The Atlantic TCs, that are generally more intense than those generated in the Caribbean Sea (Fig. 3e and f), originate high waves in the central Atlantic basin that have a long fetch before reaching the Antilles. These islands act as a barrier and prevent the large waves from entering the Caribbean Sea. After crossing the Antilles, these TCs continue travelling inside the Caribbean Sea, but the generated waves are less intense partly due to the smaller fetch area compared to their paths in the Atlantic Ocean.

The peak direction ($D_p$) of wave propagation along the Caribbean coastlines is perpendicular to the coast due to wave refraction, regardless the TCs family (Fig. 5c). The main peak period ($T_p$) found in the Caribbean region for TC-generated ocean waves (Fig. 5d) has values between 14 and 18 s. Smaller periods (around 6 s) are observed in areas with the greatest influence of TCs originating inside the Caribbean Sea (Fig. 3.1b) or in particularly protected areas such as the southern coast of Cuba or the coast of Belize, or in the enclosed bay of Haiti where its capital city, Puerto Principe, is located. Other protected

areas with low $T_p$ include Maracaibo lake in Venezuela and the Gulf of Paria in Trinidad and Tobago island, although here $H_s$ is negligible.

## 3.4 Sea surface elevation induced by TCs.

Following the same metric used for waves, the $99^{th}$ percentile of the set of the maximum SSE for each TCs at every coastal point has been used to describe the spatial distribution of extreme SSE generated by TCs (Fig. 6). Here we focus on total SSE changes and its counterparts caused by the atmospheric pressure, by winds and by wave-setup, which is the elevation of the sea surface due to breaking waves (Gregory et al., 2019). To do that, four different numerical simulations were performed for each TC as explained in section 2.3.

The largest values of coastal SSE ($> 2m$) (Fig. 6a) are found along the coastlines of Cuba and Belize, where the $H_s$ is lower (Fig. 5a), because of their shallow coastal waters that favor the wind setup. SSE values as low as 20 cm are obtained along the southern coasts of the basin, far from the main pathway of TCs (Fig. 3c). Regarding the role of each component to the total SSE, the wave-setup is negligible almost everywhere (Fig. 6a). Its contribution reaches a maximum of around 25 cm in the shallow areas of Belize, Mexico and the southern coast of Cuba, mentioned before, and it represents less than $5\%$ of the total contribution (see Fig.S2b) to the SSE. The reason is that the wave-setup is very likely underestimated due to the relatively coarse coastal resolution of the model grid that is not able to represent properly the shallow water wave dynamics. Wind-setup (Fig. 6c) is the most important contribution, specially in the western Caribbean coastlines where it can reach up to 2 m. The eastern coasts are dominated by the effects of the low atmospheric pressure associated to TCs (Fig. 6d, note the different colour-scales), with a contribution of around 0.5 m in this area. Further analysis of the link between atmospheric pressure and SSE value is done in Fig. S3. If, for each coastal point, the number of TCs at a distance less than twice the Rmax, a very similar pattern to the atmospheric pressure contribution to the total SSE emerges (Fig. 3c). Summarizing, in terms of SSE induced by TCs, the eastern Caribbean coastlines are mainly affected by the atmospheric pressure contribution, while the main driver in the western area is the wind setup. These geographic patterns are the result of the combination of two different mechanisms: 1) the crossing of TCs of Atlantic origin over the eastern coasts leaving their footprint in atmospheric pressure and 2) the shallow continental shelf along the western coasts, that allows the water to accumulate due to the effect of the wind. The effects of wave setup are not accounted for due to the model´s spatial resolution, so the numbers of SSE represent a lower bound of TC-induced coastal sea level extremes.

## 3.5 Return levels of coastal waves and sea surface elevation

Return levels for periods ranging between 1 and 500 years, specifically for 10, 50, 100, 200 and 500 years, have been computed following the method described in section 2.4. The return levels associated to the 100-year return period of $H_s$ and SSE generated by TCs are shown in Fig. 7a and b, respectively. The spatial distribution of the return levels for the $H_s$ (Fig. 7a) and SSE (Fig. 7b) mimic the geographic patterns in Fig. 5a and Fig. 6a, respectively, as expected. The 100-yr return periods of $H_s$ exceed 12 m in West Indies and most of the northern Caribbean boundary, including Puerto Rico, Dominican Republic and the eastern part of Cuba. Cuba is also affected by the largest SSE (panel b) reaching up to 2 m. High return levels of SSE are also

observed in the coast of Mexico, Belize and a small part of Guatemala. Finally, the southern Caribbean coast, once again, has
the lowest values of both $H_s$ and SSE return levels, being in some places even null, such as the coasts of Panama. These are,
therefore, the safer coastlines in the Caribbean Sea in terms of coastal hazards, as far as TCs are concerned.

## 4    Summary and Discussion

The Caribbean Sea is a region prone to the passage and impacts of TC, generated either in the Atlantic Ocean or within
the basin. Historical records of major TCs indicate that largest ones usually impact long that Atlantic-facing eastern coasts.
Hurricane Hugo, that hit Guadalupe and Puerto Rico in 1989, caused severe damages in low-lying coastal areas, both in
fatalities (56 people) and properties ($17.4 billion) (Krien et al., 2015). A more recent example is hurricane Maria in 2017, a
5-category tropical cyclone that caused almost 200 deaths (Pasch et al., 2018), although later counts reached 3000 fatalities[1];
and Hurricane Irma, that affected the same area two weeks later (Cangialosi et al., 2018), proving that the West Indies are
an oversea region that lies within the tracks of powerful hurricanes originating in the Atlantic basin. In the northern area, the
coasts of the Dominican Republic stand out as another hot spot of large wind waves induced by TCs, specially in its southern
side near the capital, Santo Domingo, with $H_s$ exceeding 14 m. This is one of the most heavily populated and densely built-up
areas of the island and, consequently, most exposed to hurricane-induced hazards. This was evidenced by Hurricane David, a
category 5 TC that hit this island in 1979 which was responsible for 2068 deaths, 2000 of which occurring in its capital (Hebert,
1980).

In this study we have analysed, for the first time, the coastal extreme sea levels generated by present-day tropical cyclones in
the Caribbean Sea by numerically simulating the sea level elevations and the wind waves from 1000 synthetic tracks selected
from STORM dataset. Two well-differentiated families of TCs have been identified depending on their generation region
(Fig. 3): the Atlantic basin family, formed by very powerful TCs that mainly affect the coasts of the Antilles, Puerto Rico,
La Española and Cuba; and the Caribbean basin family which generates off the coast of Honduras and mainly affects the
eastern coasts of the Caribbean Sea. The main differences between both families, besides their generation area, are 1) the
length of their tracks, having the Atlantic TCs longer life-times travelling longer distances before their first landfall; and 2)
their intensity (maximum wind speed), being the Atlantic TCs much more intense than the Caribbean TCs due to their longer
life-times travelling over a warm sea. The projected changes in the large-scale steering flow from increasing greenhouse gases
are expected to change their genesis location and shift their tracks (Colbert et al., 2013). These changes would affect both
families differently, declining the frequency of Caribbean hurricanes and increasing those of the Atlantic basin. For the latter,
an eastward shift on their tracks is also expected, what would translate into a decrease in the number of Atlantic TCs affecting
the Caribbean coasts. However, due to an increase in the Sea Surface Temperature (SST), the Hurricane season will, most
likely, extended, therefore a higher number of TCs could potentially affect these areas (Bustos Usta and Torres Parra, 2021).

Our results show that the Atlantic-facing eastern coast of the Caribbean basin is the one affected by the largest waves
generated by TCs (Fig. 5), easily exceeding 10 m of $H_s$ and being especially large in the Lesser Antilles (Barbados, Saint

---

[1]https://www.nytimes.com/2018/08/28/us/puerto-rico-hurricane-maria-deaths.html

Lucia, Martinique and Dominica), where $H_s$ can reach 16 m, corresponding to a 50-years return period. Previous studies, limited to some sectors of the Caribbean coastlines, have also identified other regions as heavily affected by the wind-waves generated by TCs. For example, Pillet et al. (2019a) identified as one of the most vulnerable areas the east coast of the West Indies, affected by $H_s$ larger than 15 m, and Montoya et al. (2018) pointed to the northern area of the Caribbean sea, near Jamaica, Puerto Rico, Dominican Republic and Haiti, as the regions inside the Caribbean Sea with largest 30-year return-level $H_s$ of up to 12 m. These are in agreement with our findings. Ignoring possible changes in the tracks of hurricanes affecting the Caribbean basin (Colbert et al., 2013), changes in $H_s$ from TC intensification due to rising sea surface temperatures are expected to be minor, with the largest increase not exceeding 25 cm in the Colombian Basin (Kleptsova et al., 2021). As a result patterns of $H_s$ would remain similar to those seen in Fig. 5 and 7. However, this is not the case if the effects of the projected increase in intensity of such hurricanes are considered (Murakami et al., 2014), as these could result in higher return levels than those calculated.

The largest sea surface elevations generated by TCs were found along the northwestern area of the Caribbean Sea (Cuba, Mexico and Belize) in agreement with Torres and Tsimplis (2014). Note that this study includes the contribution from tide, eddies and seasonal cycle, while here only the hydrodynamic response is considered. SSE is very dependent on the morphology of the coastlines, which in this case is formed by very narrow and shallow bays located behind cays that allow the generation of storm surges larger than 2 m, provided that the shelf is well represented in the model bathymetry. Indeed, the largest storm surges registered in the Caribbean Sea are in the Gulf of Batabanó (Cuba) with a value of more than 2 m (Arenal et al., 1998, 2016). On the contrary, the TC-induced SSE in the Antilles, the area affected by the most intense TCs, is not among the largest simulated ($< 0.5$ m). In this case, the lack of a continental shelf or narrow and shallow bays behind cays in front of the coastlines do not allow the accumulation of water having as a result smaller SSEs than other regions affected by less intense TCs. Another SSE hot spot is located in the northern coast of Nicaragua with values exceeding 0.8 m. This section of coastline, that is mainly affected by the less intense Caribbean TCs, has a wide and shallow continental shelf, prone to the development of storm surges. The smaller SSE values were found in the southern coastlines of the Caribbean basin because this region is not affected by TCs at all (Fig. 3). Higher return levels of SSE are also observed along the coasts of Mexico and Belize as well as some parts of Guatemala, with values that agree with those found by Dullaart et al. (2021a). Despite Dullaart et al. (2021a) includes tides, these are almost negligible in this area, so our return periods are directly comparable. All these zones are flooding hot areas where, in concordance with Dullaart et al. (2021b), the number of people exposed to a flood event relative to the global population due to TCs storm surges contribution is especially large ($> 90\%$), both in Cuba and Belize. It is worth noticing that the SSE in the Yucatan Channel, between Mexico and Cuba, may be overestimated because it is part of one of the boundaries of the model.

We have also investigated the contribution of each mechanism (atmospheric pressure, wind setup and wave setup) to the total SSE, which strongly varies between different regions of the Caribbean Sea. The largest contribution of the atmospheric pressure is found in areas located along the TCs tracks. The wind contribution is strongly related to the morphology of the coast and continental shelves. Wave setup contributions are also located in shallow areas that allow the accumulation of water. However, this contribution is underestimated due to the relatively coarse resolution of the model grid that prevents reproducing

properly the wave transformation in shallow waters. Mean sea-level rise will increase the water depths and is likely to cause a decrease in the contribution of wind setup and wave setup (Kleptsova et al., 2021) while keeping the atmospheric pressure contribution equal. Despite this decrease in the contribution of the wind setup, mean sea-level rise is still responsible for the positive trends observed in sea level extremes (Torres and Tsimplis, 2014).

320    This study provides a regional overview of the coastal hazards induced by TC in the Caribbean Sea. The model simulations provide a complete mapping of the extreme wind-waves and storm surges along the coasts and identifies the dominant mechanisms as well as the type of TC (in terms of their origin) and the areas that are most affected by each of them along with return levels for 10, 20, 50, 100, 200, 500-years return periods. Limitations of our study include: 1) the fact that our forcing fields have been generated using synthetic tracks and using parameterised pressure and wind profiles possibly leading to inaccurate

2D fields; 2) low bathymetry resolution which causes 3) underestimated wave setup; and 4) a possible SSE overestimation may be taking place in the Yucatán Channel since is part of one of the open boundaries of the model.

*Data availability.*    The return levels of $H_s$ and SSE corresponding to 10, 50, 100, 200 and 500 years along all coastal grid points, along with the data (latitude, longitude, Rmax, minimum pressure and maximum wind speed) for the TC's subsample, and the results (Maximum of SSE and Hs, Median of Tp and Dp) for all simulations are available at the data repository 10.5281/zenodo.7069110

*Author contributions.*    AA and MM conceived the work and designed the numerical experiments. AM retrieved the atmospheric forcings and performed the numerical simulations. AA, AM and MM analysed the outputs and all the authors contributed to the outline and writing of the manuscript.

*Competing interests.*    The authors declare that they have no conflict of interest.

*Acknowledgements.*    Grant PID2021-124085OB-I00 funded by MCIN/AEI/10.13039/501100011033/FEDER,EU and grant RTI2018-093941-

B-C31 funded by MCIN/AEI/ 10.13039/501100011033 and by "ERDF A way of making Europe".

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

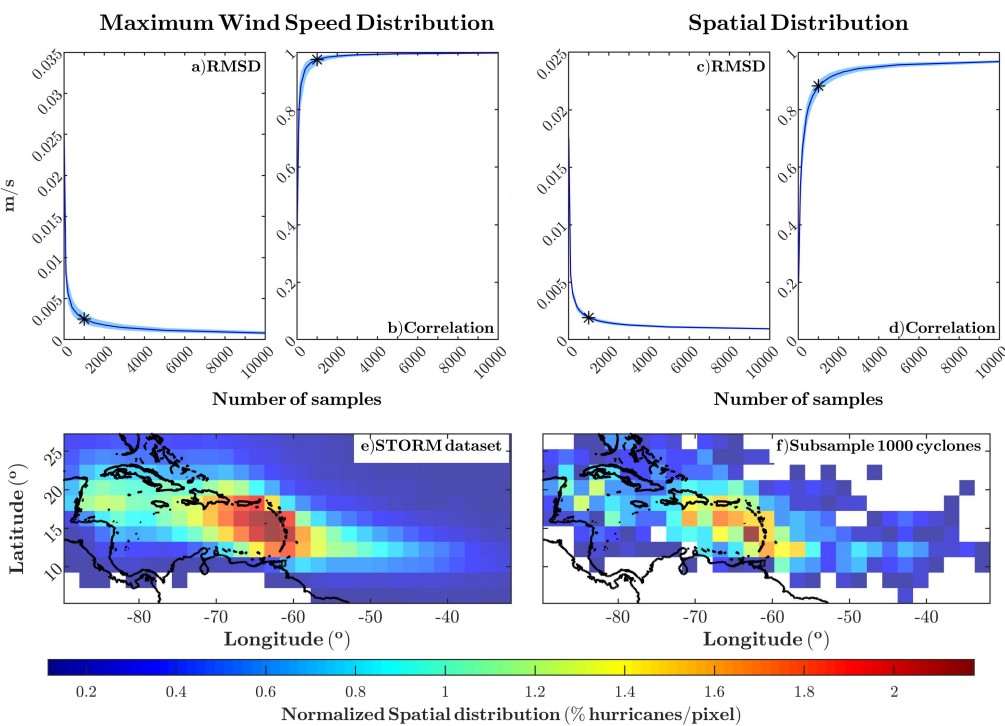

**Figure 1.** Statistical tests carried out on the Caribbean cyclone subsamples using as variables: the maximum speed for the panels a), b), and the spatial distribution for the panels c) and d). The blue line represents the median and the light blue part represents the range of values between the percentiles 95 and 5, on each panel. Units of panel c) are the normalized number of TC/pixel, where the normalization is defined as the number of TC affecting each pixel divided by the total. The panels e) and f) represent, respectively, the the percentage of cyclones per pixel for the total sample ($\sim$ 25000 TC) and the subsample selected for the project ($\sim$ 1000 TC), marked on the rest of the panels.

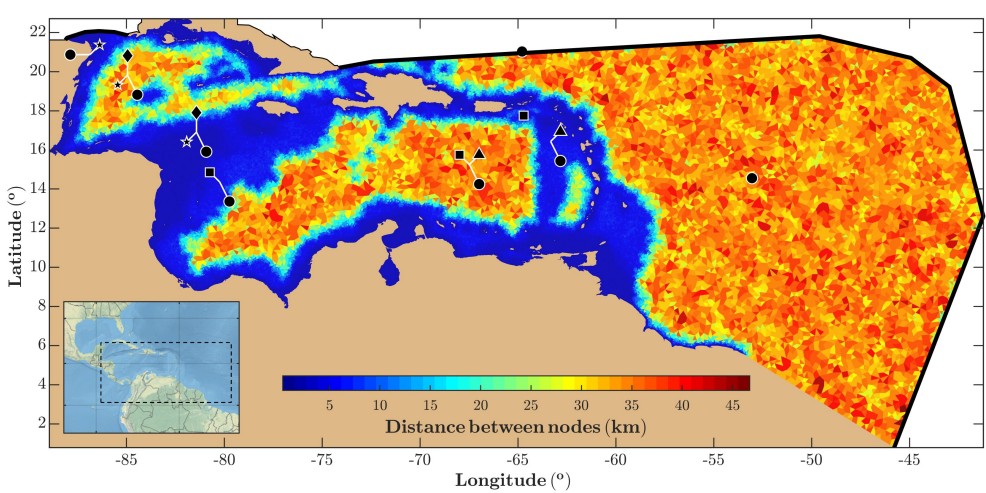

**Figure 2.** Domain of simulation (with the inset indicating the global region) and the computational grid. The colours represent the distance between nodes in the computational grid, with the spatial resolution varying with depth. Black thick lines represent the open boundaries of the model domain. Black markers represent the location of buoys and tide gauges used for the validation, with the shape indicating the corresponding TC, as in Table 1. In some cases, the same instrument is used for various cases, with the exact position denoted by white lines.

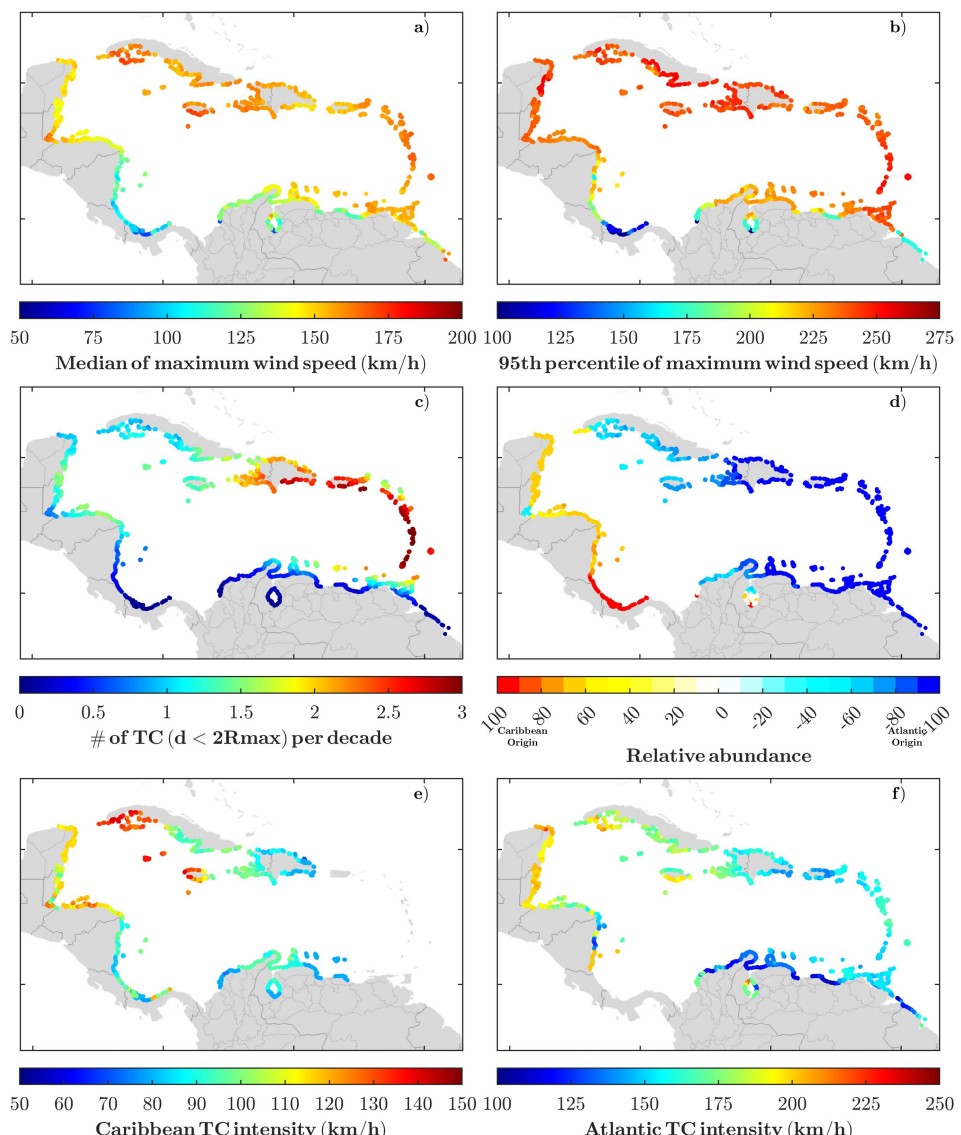

**Figure 3.** Characteristics of TCs affecting the Caribbean coasts. The top panels a) and b) show the intensity of the selected subset of TC in terms of the mean and the $95_{th}$ percentile of the maximum wind speed when they reach the coastline. Panel c) the number of TCs per decade affecting each coastline, considering that a coastal grid point is affected if is within a radius smaller than twice Rmax. Panel d) represents the family of TCs (Atlantic or Caribbean) that most affect every grid point, in percentage. In this way, 0 is obtained if both families affect equally, and in colour the percentage of the one that affects more. The entire red zone is dominated by TCs generated in the interior of the Caribbean Sea, while the blue zone is more influenced by TCs of Atlantic basin. Finally, panels e) and f) show the median maximum wind speed of TCs from each family.

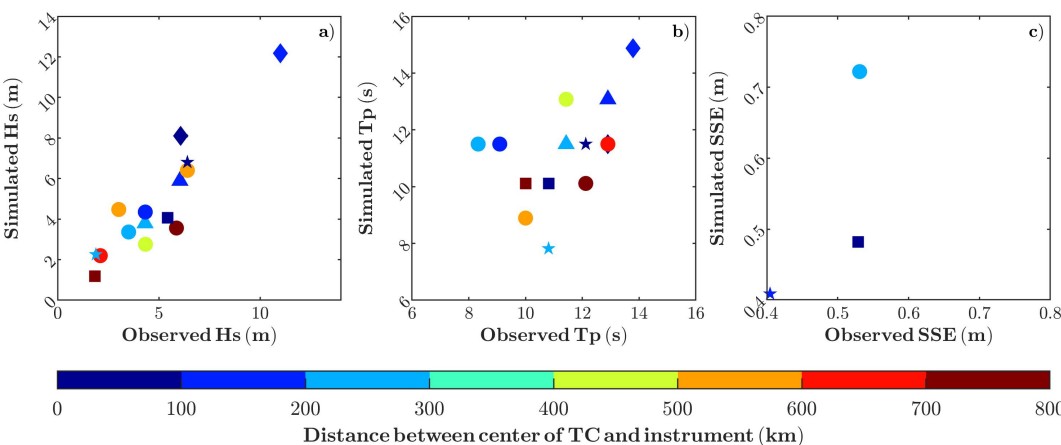

**Figure 4.** Comparison between modelled $H_s$, panel a), $T_p$, at the maximum of $H_s$, panel b) and SSE, panel c) with the observations from in-situ buoys (a,b) and tide gauges (c). The forms indicate the TC (see Table 1) and the colours correspond to the distances between the eye of the TC and each instrument at the moment of largest observed value (see Fig.2).

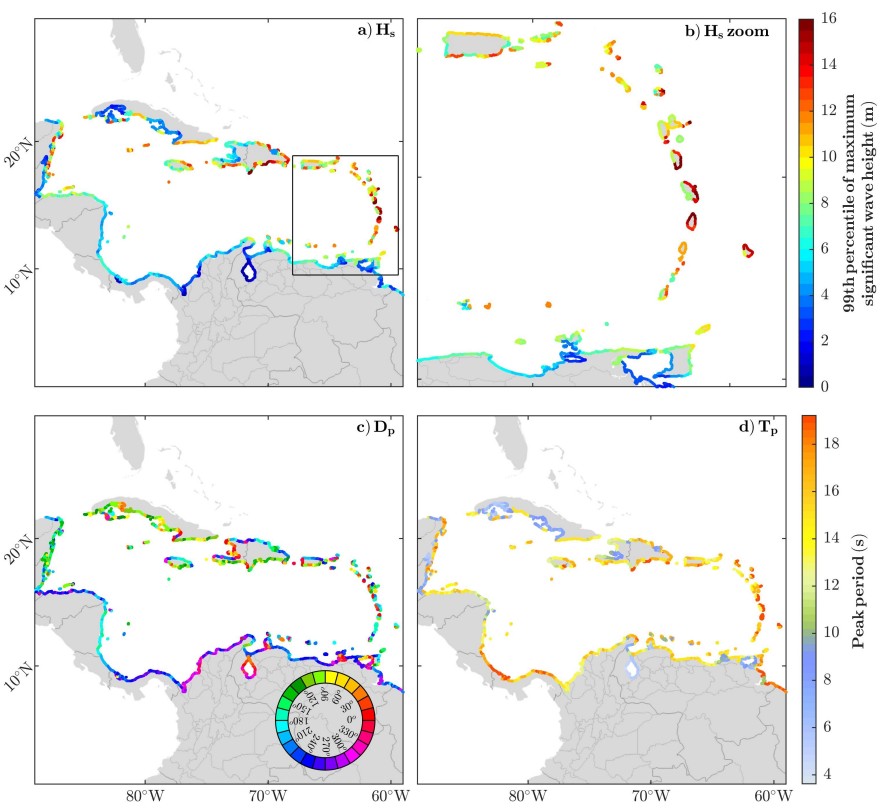

**Figure 5.** Fully coupled simulation results. Both panels a) and b) represent the $99^{th}$ percentile of the maximum $H_s$ (that is, the value corresponding to the 10 most intense TC), for the whole domain (a) and a zoom on the Antilles area (b). Panels c) and d) are the peak direction ($D_p$), in nautical convention, and the median of peak period ($T_p$) for waves over the $99^{th}$ percentile, respectively. All panels were calculated using the nearest point to coast at a depth of 20 m, to avoid problems with relatively coarse shore resolution.

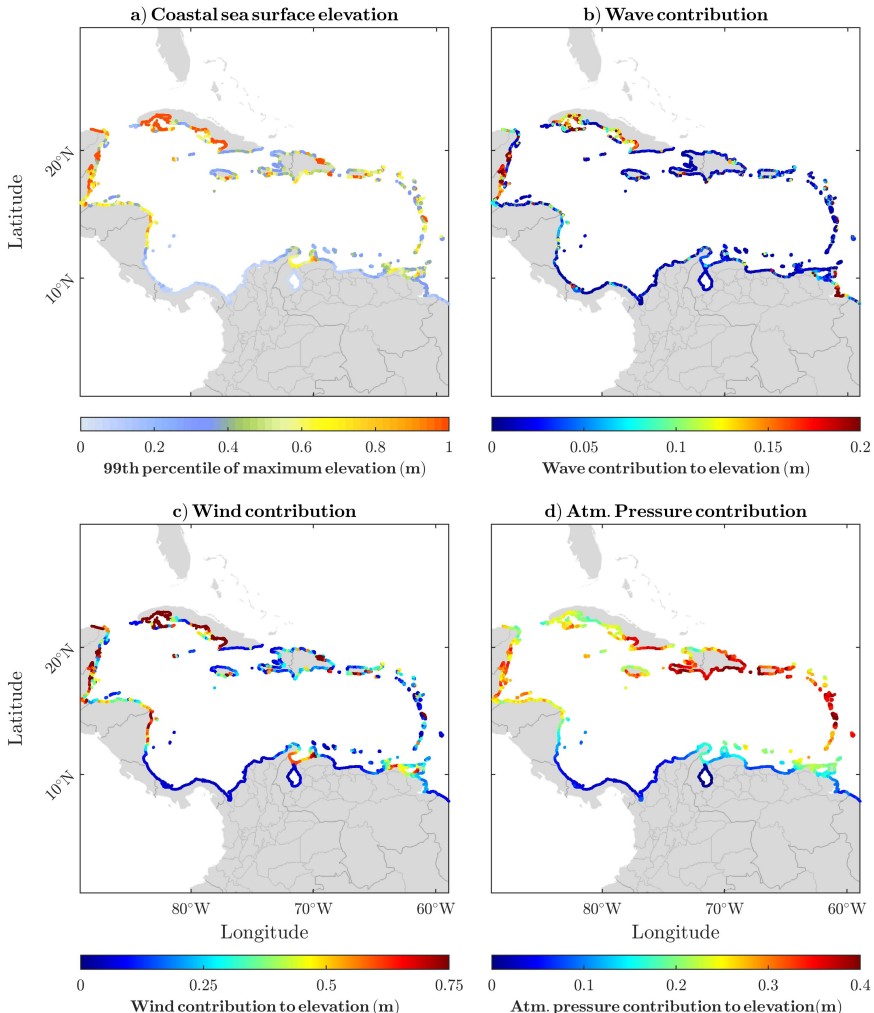

**Figure 6.** $99^{th}$ percentile of the maximum sea surface elevation (a value that corresponds to the 10 most intense TCs at each grid point) along the Caribbean coast (a) and its contributions: wave setup (b), wind (c) and atmospheric pressure (d) in absolute terms. All panels were calculated using the nearest point to coast at a depth of 20 m, to avoid problems with relatively coarse shore resolution.

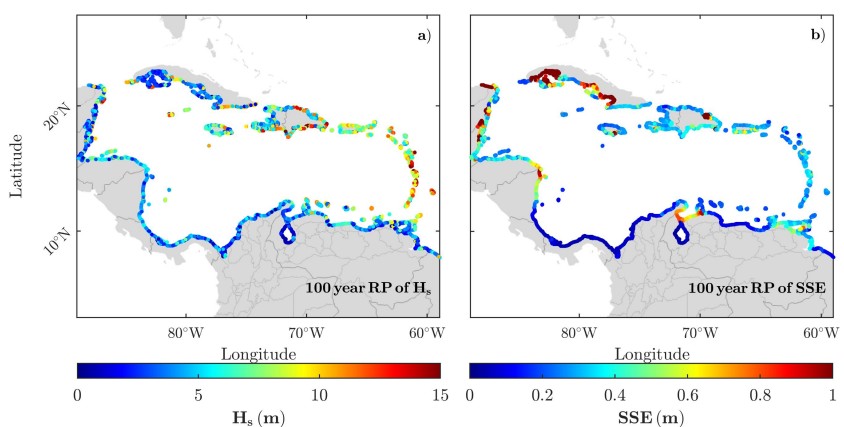

**Figure 7.** Return levels of the variables $H_s$ in panel a), and SSE panel b) for 100 years return period.