# Peer review of "Coastal extreme sea levels in the Caribbean Sea induced by tropical cyclones"

_Natural Hazards and Earth System Sciences, 2022_

## Referee Comment (RC2)

**REVIEW, "COASTAL EXTREME SEA LEVELS ..."**

MARTÍN, AMORES, ORFILA, TOOMEY, MARCOS

In their manuscript "Coastal extreme sea levels in the Caribbean Sea induced by tropical cyclones," the authors Martín *et al.* use a new database of synthetic tropical cyclones as forcing to simulations of wind waves and storm surges in the Caribbean Sea. It is shown that the wind waves and storm surges vary significantly at coasts around the basin, due to differences in storm evolution, local bathymetry, and other characteristics. The manuscript is generally well-written.

It is my recommendation that this manuscript can be **rejected** for publication.

To this reviewer, there are two critical flaws in this study. First, the validation is inconsistent with the rest of the study and is inconclusive about the quality of the model predictions. The two validation storms are represented by data-assimilated atmospheric products, whereas the synthetic storms are constructed from a simple parametric model. And the two validation storms are (apparently) not described by sufficient observations of wind waves and coastal sea levels, so it is not possible to validate the model predictions for these storms. Without a validation, it is not possible to trust the later predictions for the synthetic storms – this is recognized by the authors, who note an insufficient nearshore resolution and possible boundary effects as reasons for poor predictions of wind waves and storm surge, respectively. The validation should be expanded to investigate these and other potential problems, either by validating against all available observations for these storms, or by selecting other storms with useful observations.

Second, the findings are not necessarily novel, and it is not clear what is the contribution to our scientific understanding. Previous studies have investigated storm-induced hazards in the Caribbean Sea and elsewhere, and they have characterized the wind waves and storm surge that is possible along their coastlines, as well as the relative roles of atmospheric pressures, winds, and waves as drivers. What are the gaps in those previous studies, not in number of storms considered, but in understanding of coastal processes? How will this study help to fill those gaps? As-is, the study is impressive in the amount of computations that have been performed, but it is lacking in connecting those computations to a novel contribution. Because these two flaws will require a substantial refocusing of the manuscript, likely with additional computations, this reviewer recommends the present manuscript to be rejected for publication.

The following major comments can be considered in a revised manuscript:

- Lines 25–37: It is not clear (at least to this reviewer) what will be learned by reading this manuscript. It is stated that "we focus on the ocean hazards generated by TCs in terms of wind-waves and storm surges." Have these hazards not been characterized previously,

either basin-wide or at specific locations in the Caribbean Sea? Do we expect this 1000-storm study to provide new insights into the magnitudes of waves and surges in this region? If so, then why?

It is also stated that "[w]e analyse in detail the outputs of the numerical simulations to quantify the role of the different forcing factors." Have these roles not been quantified previously? This reviewer is aware of several studies that quantified the relative roles of atmospheric pressures, winds, and waves on the magnitude of storm surge. Do we expect this 1000-storm study to provide new insights? If so, then why?

The Introduction would benefit from clear questions and hypotheses to motivate the study.

– Lines 49–63: Can the authors provide more justification for how the storms were selected? This reviewer is interested in two aspects. First, why use the maximum wind speed (presumably, see minor comment below) as a proxy for the storm intensity? Why not use the minimum central pressure, or maximum radius, or an integrated quantity like the total kinetic energy? The STORM database includes several parameters for each storm, not just the maximum wind speed, but it seems like the current methodology is ignoring them.

Second, the random selection of subsamples seems suboptimal. Why not use a maximum dissimilarity algorithm to identify the top 1000 storms that best span the parameter space? Or surely there are other methods that could be considered? The authors' method appears to work okay, given the convergent errors shown in Figure 1, but it would be nice to see a brief discussion of why this method was selected over other options.

– Lines 79–81 and 170–173: Not sure what this means. There were 46 sea-level records in the region – they must have observed something useful about the water levels. The authors refer to a lack of "footprint" – does this mean that the observation records do not include the effects of the storms? The lack of water-level validation is a critical flaw in this study. Somehow the water-level predictions need to be validated, either for these or other storms. Without a validation of the water levels, the rest of the results cannot be trusted.

– Lines 92–94: References are needed for these methods. Why use a distance of 300 km? Why use a reduction of 20 percent?

– Sections 2.1 and 2.2: There is a mismatch between the atmospheric forcing used for the validation and the rest of the study. The validation storms use a data-assimilated product, which should be accurate (although this reviewer is not convinced that ERA5 can resolve the full dynamics of a tropical cyclone), whereas the synthetic storms are developed with a parametric model. The validation storms have different resolution (1 hr, 30 km) than the synthetic storms (3 hr, 5 km). Can the authors comment on how these differences may affect the validation?

More importantly, why not generate the atmospheric forcing for Wilma and Tomas in the same way as the synthetic storms? The authors could use the historical track information for these two storms, push it through the parametric model, and then be able to compare apples-to-apples.

– Figure 4: Why use only two buoys per storm? Why not do a comprehensive validation by using all available observations? As-is, the reader can assume that these buoys were cherry-picked to show the best results.

– Lines 184–185: "but the generated waves are less intense due to the smaller fetch area." The Caribbean Sea is a large basin, with a minimum width of 600 km at its narrowest. Why would any waves be fetch-limited in this basin?

– Lines 204–205 and 294–295: This claim should be explored, ideally via a more-comprehensive validation with the full set of available observations. But more importantly, why do the authors think the wave set-up is under-estimated? Should it be more than 5 percent of the total contribution?

– Lines 305–306: The repository should include more than just the return periods. The selected 1000 synthetic storms should be included, both in their parameters from the STORM dataset and their pressure/wind fields from the parametric model. The SCHISM and WWM-III input files should also be included. This will allow for reproducibility of the study results.

– Section 4: An overarching comment is that it is not clear (at least to this reviewer) what is novel about the study findings. It should be expected that the windward islands are affected by Atlantic storms, whereas the west side of the Caribbean is affected by storms from that basin. It should also be expected that regions with narrow shelfs and deep offshore bathymetry will have smaller storm surges that are forced mainly by the storm's pressure deficit, whereas regions with wide shelfs and shallow offshore bathymetry will have larger storm surges that are forced mainly by the storm's winds. Can the authors do more to contextualize their findings and motivate their novelty?

The following minor comments can also be considered:

– Line 57: "trough" should be 'through' for correct spelling.

– Figure 1: It would be helpful to describe what is meant by "speed." This reviewer assumes it is the maximum wind speed at any location/time during the storm. But it could be something else, e.g. the forward speed of the storm.

– Line 65: Again, it is assumed that these speeds (e.g. 111 km/h) refer to the maximum wind speeds, but this should be clarified in the text.

– Figure 2: This reviewer struggled to see the tracks and labels in this figure, as they were depicted in black on a mostly blue background. Not sure what to suggest to make these features to be more legible. What if the track and labels were in white?

– Line 78: Should be 'data were' for subject-verb agreement.

– Line 97: "last" should likely be 'latest.' Please give the actual version numbers for SCHISM and WWM-III.

– Line 122: "where" should be 'were' for correct spelling.

– Line 275: "are" should be 'area' for correct spelling.

– Line 280: When the letter 'm' is shown in italic font, this reviewer assumes it is a variable, e.g. 25 times m. If it is meant to be a unit (meters), then it should not be in italic font.

---

## Author Comment (AC1)

Response to Reviewer #1 of our manuscript entitled

**Coastal extreme sea levels in the Caribbean Sea induced by tropical cyclones submitted to *Natural Hazards and Earth System Sciences.**

Ariadna Martín, Angel Amores, Alejandro Orfila, Tim Toomey, Marta Marcos

October 7, 2022

The paper uses a synthetic hurricane database to obtain a set of 1000 tropical cyclones (TC) affecting the Caribbean Sea. This information is used to assess wind speed, wave height and extreme sea levels in the region using a high-resolution coupled hydrodynamic-wave model. Results allow identifying most exposed areas to these variables depending on the origin of the TC (Caribbean or Atlantic). Besides, wave, wind and atmospheric contributions to the extreme sea levels are assessed.

I find the aim of the paper interesting and scientifically relevant, especially for the use of coupled models to simulate a large number of present storm surges and waves in the Caribbean Sea forced by TC. I believe this is important as many areas in this region, lack of observational wave and sea level data to perform TC-related risk assessments. Therefore, these results are useful to assess coastal TC risks in the entire basin, but especially at places were data is not available. Besides, return levels of wave height and sea surface elevation for the Caribbean coasts are available in a data repository.

The paper has a good presentation quality and falls into the NHESS scope. However, in my opinion some aspects need to be improved before considering its publication in the Natural Hazards and Earth System Sciences journal.

Main comment:

(1) — Buoy data from two real hurricanes (Willma and Tomas) are used to validate the modelled wave height, but authors claim that no sea level data is available to validate the sea surface elevation (SSE, L171-172). This is not accurate. In fact, Figure 7 from a referenced paper (Torres and Tsimplis, 2014) shows the nontidal storm surge produced by Hurricane David (1979) as recorded by the Magueyes tide gauge. Besides, this hurricane is mentioned later in L262. In the context of the paper, any hurricane in the region could be used for validation purposes. I recommend the authors to select two hurricanes with available sea level and wave observed records, as these variables validation is important for this research. Besides, as the paper assess hurricanes formed in the Atlantic and in the Caribbean, I think it would be better to validate the model with two real hurricanes but from different origin.

A — We thank the reviewer for the information provided. Following this and other requests, we have completely changed the validation section. Now we include all TCs for which there is information on either tide gauges or buoys after year 2000. We need to restrict ourselv to this period of time because we require information on the radius of the TC to be able to convert it to 2D fields. Please, see the new section for further explanations.

Besides, we have checked the results with hurricane David. It generated a signal in the tide gauge of Magueyes, however, the winds provided by ERA5 in that area are very underestimated (See Fig. 2a)). For both TC David and Tomas (Fig. 2 and 1 a, note that the numbering refers to the present document), winds from ERA5 are underestimated near the track, that is mainly because of the size of the TC, compared to the resolution of ERA5 [Dullaart et al., 2020]. For a tide gauge to pick up an important signal, th TC has to pass very close to the tide gauge, making it very difficult to have any

good reanalyses in the area. The surge effect can be be felt at a greater distance from the center of the hurricane, where winds of ERA5 are perfectly consistent with the observations (see 1b) ). That would be the reason why the validation of sse in Fig. 2 b) is underestimated. Another simulation can be made by calculating the forcing fields using the date from IBTrACS, (See Fig. 1 b and c), however, this can not be made for TC David because IBTrACS has not available data of the radius of this TC.

However, we have found some other tide gauges and we were able to remake the validation section. We have changed Sec.3.2 for this purpose.

Other specific comments:

**(1) — L43. Please clarify if tropical cyclones from STORM database, include Tropical Depressions and Tropical Storms in addition to Hurricanes of all categories. If this is the case, the term "tropical cyclones" should not be replaced for "hurricanes".**

A — Thank you for pointing this out. The STORM dataset is made by selecting a threshold of 18 m/s, which corresponds to a Tropical Storm classification. Therefore, we have changed all corresponding "hurricanes" with the term "Tropical cyclones" or TCs.

**(2) — L56. Maximum wind speed.**

A — Corrected

**(3) — L57. Please clarify how the PDF was built. The dataset provide the TC track with a 3-hourly time step during its lifetime. For the PDF (and figures 1e and f) each TC was counted once at the time of its maximum wind speed or the entire track was used?**

A — Each TC was counted at every time step. This is because, to compute the spatial correlation, we want to make sure that not only the genesis of the TCs is well selected but the tracks are consistent, too. Therefore, we count every time step of the track, to identify those places where there is a higher concentration of hurricanes affecting that area. We have added this explanation to the text.

**(4) — L 63. Please expand on the characteristics of the 1000 TC sub-set used. For example, indicate the % of TC with Atlantic or Caribbean origin, % or number of TC which reached the different hurricanes categories (for each origin), etc. This information might be useful to assess the results. Besides, explain how the TC were classified due to their origin, as this is discussed in the paper. E.g. in the case of TC of "Caribbean Origin", are they included if the track starts in this region (usually as a tropical depression), or depending on the place where the TC became a hurricane? (This is comment is related to comment about L43).**

A — Please, note that this comment has been addressed in section 3.1. We have included the % of TCs of different origin, among those that affect the region and in the figure we added two panels to show the mean intensity and maximum wind speed reaching every coastal point (also in response to request 10 below). We do not include the categories, though, as the maximum could be reached outside the domain and would in that case be meaningless in terms of impacts.

Regarding the classification into families of TCs, it is specified that their region of origin is used, irrespective of whether they become TCs (see first paragraph Sec. 3.1, prior to percentage information)

**(5) — L83. Please clarify the area where the 47 buoys were available, as the reader can understand that these buoys were available in the Caribbean Sea.**

A — There are 34 buoys available inside the area of study, 27 of which are found inside the Caribbean sea, and 7 in the nearby Atlantic Ocean. This information has been added to the text (section 2.1).

**(6) — L87. I suggest you clarify that "... surface wind fields for each TC were generated ...".**

A — Done

**(7) — L89. Clarify the term "dominant circular component". Replace it for "a circular region affected by the cyclonic wind" or a similar expression that better explain the wind field**

configuration.

    A — Done

**(8) — L94. Include a reference or explain why a 20% is a good choice.**

    A — Done, we have included the reference [Willoughby and Black, 1996]. We would like to note, however, that this is not really a relevant factor, as those winds do not generate waves or storm surges, once the TC has made landfall.

**(9) — L144-145. Please consider including a comment about the latitude of the southern Caribbean and its relation to the weak Coriolis Effect, which affects TC.**

    A — We are unsure about what the reviewer requests here.

**(10) — L155. A comment about intensities from both TC's families is presented based on Figure 3. Panels c) and d) in this figure are interesting as they show the coastal areas where strong TC induced winds are expected depending on their origin. Please consider to complement this figure with a panel of intensity (median of maximum wind speed) of all the cyclones regardless of their origin, as well as a second panel but with the intensity as the 95 percentile of maximum wind speed. This information can be useful for coastal planning and risk management, as coastal infrastructure should be prepared for strong TC winds regardless of the cyclone's origin. Include in the text an assessment based on these new panels.**

    A — Done. See Fig. 3 panels a) and b).

**(11) — L161. generated in the Caribbean; eastward; westward (check all the manuscript).**

    A — The answer is in the next questions

**(12) — L162-163. The STORM database include 20 TC moving eastward. Although this is a very small number, they seem to be strong hurricanes when compared to other cyclones formed inside the Caribbean Sea (Fig. S1c). Is there a real hurricane that have shown this behavior? I think this is important in order to mention if this is a real possibility in an area dominated by north Trade winds or if this might be catalogued as an error in the database.**

    A — The reality is that there may be TCs generated in the interior of the Caribbean whose trajectory ends up affecting the Antilles. However, these are very rare. Moreover, they are not part of the Caribbean family to which we refer, whose genesis is off the coast of Honduras, but are hurricanes that are generated directly on the coasts of the Antilles and then continue northward. We have not discussed this subgroup in detail since it is not an error in the database, but a very rare family of events. In STORM they correspond to less than 0.001% of the total number of TCs generated in the Caribbean Sea

**(13) — L172. To assess the hurricane effects on sea level, high frequency data is needed, which is not available from altimetry. Therefore, I recommend to delete this comment.**

    A — Done

**(14) — L176. Replace ten for tenth if it uses Hs from the TC in the tenth position (99th percentile). If Hs is computed form the ten most intense hurricanes, which is the "measure" (¿mean, median?). Please clarify the method used in this line for Hs and in L194 for SSE. Based on this clarification, update figure 5 and 6 legends.**

    A — We use maximum $H_s$ for each TC modelled at every coastal grid point to build a time series and then we keep the $99^{th}$ percentile. So, yes, the reviewer is right and we measure the signature of the 10 most intense TC reaching the coasts and we have modified the text as suggested.

**(15) — L178. Wilma Cat 5 Hurricane recorded an 11 m Hs, while Tomas Cat 2 hurricane, recorded a 6 m Hs (Fig. 4). I think waves of almost 20 m are too high for the Caribbean Sea. I suggest you check Hs in the long time series available from the buoys used in Fig. 4, to assess**

the highest wave height recorded (and peak period – L188), and compare to the ∼20 m value found with the model.

A — Please, see the answer in the next question.

(16) — L178. Results indicate waves of nearly 20 m height in the West Indies eastern side induced by TC's. These results call my attention as major hurricanes, which can produce such large waves, are uncommon in the West Indies (e.g., https://www.nhc.noaa.gov/climo/images/1851_2017_allstorms.jpg). Therefore, validating the model with a large hurricane from Atlantic origin is recommended (see main comment).

A — Waves of more than 15 m correspond to return periods greater than 100 years (see Fig. 7a). Other comparisons with observations and other studies are included in the discussion (see Sec.4 from L: 252-274). We would like to note that all are in agreement with our study.

(17) — L185. The small fetch area inside the Caribbean Sea is proposed as the reason for smaller TC's induced waves in the western Caribbean. ¿How does the wave model forced by the hurricane wind field account for the fetch? I think wave height produced by a hurricane is related to the wind intensity, radius of maximum wind speed and the period this wind is transferring momentum to the sea surface, all of these variables forcing the model. Please consider which of the forcing variables of the wave model can be responsible for the smaller wave height seen in the western Caribbean.

A — The idea we are trying to transmit is that, considering two identical TCs originated inside and outside the Caribbean Sea, the one generated outside has a larger area of development and intensify. That would be the reason why more intense hurricanes generated in front the coasts of Honduras generate lower waves when arrive to the south coasts of Cuba, however, they can still intensify in their way to the coasts of Mexico and Florida. Then, there is another effect of the Lesser Antilles in the already develop TCs from the Atlantic, as they act as a barrier that protect the inside of the Caribbean coasts.

(18) — L188. I think waves of 14 to 18 s period are too long to occur inside the Caribbean Sea. I suggest comparing these values with Tp recorded by the buoys and/or make a comment about the period in the validation section.

A — We have compared the outputs of the model, also the period, and it seems to show good results in the validation. Therefore, despite the large values for the periods we see no reason to believe that the results provided are incorrect. We show you the validation of this $T_p$ for the same buoys and TCs used for the validation of $H_s$ in Fig. 3 here in the review.

(19) — L199-200. I expected such behavior in the coast of Nicaragua, due to the large continental platform, but Belize bathymetry is not particularly shallow (Fig 2). In the case of Nicaragua I think this research is very useful, as to my knowledge there are no observed wave or sea level data in this region, what makes validation impossible. Can you think of a reason why wind setup does not cause higher SSE in Nicaragua when compared to e.g., Belize?

A — There is a very small percentage of TCs affecting the southern coast of Nicaragua. It also has a continental shelf smaller than the norther part, which as we know makes it difficult to generate SSE due to its less local effects (wind and wave-setup). The case of Belize if special due to the morphology of the its coasts full of very narrow and shallow cays.

(20) — L204. Wave-setup is underestimated in your results. Through a numerical experiment (running a case with better spatial resolution in the model) can it be estimated the SSE underestimation? Besides, SSE validation can give an idea of the underestimation value (main comment).

A — Thank you for the comment but it is beyond of the scope of the present manuscript. We are doing this computation in the Mediterranean Sea and requires a lot of computational effort.

(21) — L237. Summary and Discussion. I suggest rearranging this section as follows: 1) summary of the study. 2) Assessment of TC intensity and Hs results. 3) Assessment of SSE results. 4)

Comparison of results with historical hurricanes as impacts are due to wind speed, Hs and SSE interaction. 5) Global warming probable effects in the Caribbean future hurricanes. 6) Closing paragraph.

A — Our section is structured as follow: 1) Brief explanation of the study and characteristic of the TCs affecting our area of study 2) Discussion of the $H_s$ 3) Discussion of the SSE along with their components, all accompanied by comparisons with other studies and possible future changes due to climate change. Finally ends with a paragraph of limitations. We believe this structure is easy to follow and understand, therefore no change have been made. However, if the reviewer thinks otherwise we can arrange some changes.

**(22) — L242. Hurricanes from Caribbean origin were only formed off the coast of Honduras? See comment of L63.**

A — See response in reviewer's comment on line 63.

**(23) — L266. Montoya et al., (2018).**

A — Done.

**(24) — L269. Consider to mention that, due to SST increase, the hurricane season will be probably extended. Doi: 10.1007/s10236-021-01462-z**

A — Done. (Lines 254-255)

**(25) — L278. Make a comment about the differences between observed sea level extremes and SSE found in this work. The former will include the contribution from the tide, eddies and seasonal cycle, while the later only accounts for the TC forcing.**

A — Done.

**(26) — L295. Mean sea-level rise might cause a decrease in the contribution of wind setup, but it is responsible for positive trends observed in sea level extremes (Torres and Tsimplis, 2014).**

A — Added

**(27) — L313. Include all available DOI to the references.**

A — Done

Comments to figures:

**(28) — Figure 1. The last panels are e) and f) no f) and g). Update the legend based on comment about L57.**

A — Done

**(29) — Figure 2. Include in the legend "The spatial resolution varies as a function of the depth". Name first Wilma (a,b) and then Tomas (c,d).**

A — Done

**(30) — Figure 3. (i) I do not understand why some segments of the coast appear with no color, as the colorbar starts from zero (e.g., Darien Gulf). Besides, it is curious that in some coastal segments no color appear, while neighbor coasts show that are regularly affected by hurricanes. E.g., a segment of the southern coast of Dominican Republic. This seems to be a technical fault, as information about this segment is available in other figures. (ii) In panel b) I understand that a 20% of Caribbean origin (light yellow) will indicate that of 10 TC, 2 more (20%) TC were originated in the Caribbean; therefore 7 TC are of Caribbean origin while 3 TC are of Atlantic origin. I think this color scale is confusing for only two possible outputs. I recommend reporting only the % of Caribbean origin TC using a colorbar from 0 (blue) to 100% (red). Therefore, 70% will indicate the percentage of Caribbean origin, what easily indicates that 30% are from**

Atlantic origin. Besides, no color will indicate areas where TC do not affect the coastline. (iii) See comment about L155.

A — Now the panels have changed as you request in question 10. Regarding the doubts of each panel: (i) Uncoloured places indicate that there are no TCs affecting those areas. As can be seen later (see Fig.5 and 6), these same areas will be areas of null $H_s$. (ii) We have tried to represent as much information as possible condensed in a single panel so as not to have to duplicate the figures, especially now that we have added 2 extra panels to the figure. If we only put the total % of one of the families, then the areas without colour could indicate that they are affected by the other family, or not affected by any of them. (iii) Done.

**(31) — Figure 4. Consider including some marks in the dates to indicate the category of these hurricanes in this time span, as this would help to see the wave height relation to the hurricane category.**

A — We have changed the way we represent the validation in order to add more events to the validation. This figure is no longer in the manuscript.

**(32) — Figure 5. (i) Remove titles at the right side of the figure. (ii) Clarify the legend accordingly to comment about L176 (note that in the legend, both reported Dp and Tp are the median from a range of data). (iii) Inverse the order in the legend as panel c) is Dp. (iv) The color code of Dp is confusing. E.g. the coast of Nicaragua is blue (relative 180° shown at the left of the color circle); as wave direction is from where the wave comes, I assume that it indicates that the waves comes from the east (the opposite side of the circle). I recommend eliminating the relative degrees in the color circle, and flipping the circle, showing light blue colors at the left and yellow colors down. An explanation about the color code should be included in the figure's legend.**

A — We have clarified the legend to make it consistent with the text and the comment above. We have also corrected the order. As for the circular colourbar, it shows the $D_p$ in nautical convention, so we have added this information on the caption.

**(33) — Figure 6. Clarify the legend accordingly to comment about L176.**

A — The caption has been changed in the same way as in the previous figure.

[Figure]

Figure 1: Panel a) representes the along track winds extracted from IBTrACS and the nearest point of ERA5 to the tracks, for Hurricane Tomas. Panel b) representes the winds near one the buoys select for the validation during the pass of Hurricane Tomas, for both ERA5 reanalsys and the winds for the buoy also with the synthetic winds created from the IBTrACS dataset. For panel c) a validation fo Hs using all winds from panel b) is shown.

**References**

Job Dullaart, Sanne Muis, Nadia Bloemendaal, and Jeroen CJH Aerts. Advancing global storm surge modelling using the new era5 climate reanalysis. *Climate Dynamics*, 54(1):1007–1021, 2020. doi: https://doi.org/10.1007/s00382-019-05044-0.

HE Willoughby and PG Black. Hurricane andrew in florida: Dynamics of a disaster. *Bulletin of the American Meteorological Society*, 77(3):543–550, 1996. doi: https://doi.org/10.1175/1520-0477(1996)077⟨0543: HAIFDO⟩2.0.CO;2.

[Figure]

Figure 2: Panel a) representes the along track winds extracted from IBTrACS and the nearest point of ERA5 to the tracks, for Hurricane David. Panel b) shows the sse from the Magueyes tide gauge, and sse simulated using ERA5 to force the model.

[Figure]

Figure 3: Each column represents the validation for the $T_p$ using the same buoys used for the validation of $H_s$ in Fig.4, for both Hurricanes Wilma and Tomas respectively.

---

## Author Comment (AC2)

Response to Reviewer #2 of our manuscript entitled

**Coastal extreme sea levels in the Caribbean Sea induced by tropical cyclones submitted to *Natural Hazards and Earth System Sciences.**

Ariadna Martín, Angel Amores, Alejandro Orfila, Tim Toomey, Marta Marcos

October 7, 2022

In their manuscript "Coastal extreme sea levels in the Caribbean Sea induced by tropical cyclones," the authors Martín et al. use a new database of synthetic tropical cyclones as forcing to simulations of wind waves and storm surges in the Caribbean Sea. It is shown that the wind waves and storm surges vary significantly at coasts around the basin, due to differences in storm evolution, local bathymetry, and other characteristics. The manuscript is generally well-written.

It is my recommendation that this manuscript can be **rejected** for publication.

To this reviewer, there are two critical flaws in this study. First, the validation is inconsistent with the rest of the study and is inconclusive about the quality of the model predictions. The two validation storms are represented by data-assimilated atmospheric products, whereas the synthetic storms are constructed from a simple parametric model. And the two validation storms are (apparently) not described by sufficient observations of wind waves and coastal sea levels, so it is not possible to validate the model predictions for these storms. Without a validation, it is not possible to trust the later predictions for the synthetic storms – this is recognized by the authors, who note an insufficient nearshore resolution and possible boundary effects as reasons for poor predictions of wind waves and storm surge, respectively. The validation should be expanded to investigate these and other potential problems, either by validating against all available observations for these storms, or by selecting other storms with useful observations.

Second, the findings are not necessarily novel, and it is not clear what is the contribution to our scientific understanding. Previous studies have investigated storm-induced hazards in the Caribbean Sea and elsewhere, and they have characterized the wind waves and storm surge that is possible along their coastlines, as well as the relative roles of atmospheric pressures, winds, and waves as drivers. What are the gaps in those previous studies, not in number of storms considered, but in understanding of coastal processes? How will this study help to fill those gaps? As-is, the study is impressive in the amount of computations that have been performed, but it is lacking in connecting those computations to a novel contribution. Because these two flaws will require a substantial refocusing of the manuscript, likely with additional computations, this reviewer recommends the present manuscript to be rejected for publication.

A — We thank the reviewer for the time taken to evaluate our work and for the assessment provided. We respond to all concerns below in detail. In particular, we have extended our validation by 1) including consistent comparisons using both outputs from the atmospheric reanalysis and the parametric model, 2) adding more TCs and 3) accounting for coastal sea levels. We have also highlighted the strong points in our work.

Main comments:

**(1) — Lines 25–37: It is not clear (at least to this reviewer) what will be learned by reading this manuscript. It is stated that "we focus on the ocean hazards generated by TCs in terms of wind-waves and storm surges." Have these hazards not been characterized previously either basin-wide or at specific locations in the Caribbean Sea? Do we expect this 1000- storm study to provide new insights into the magnitudes of waves and surges in this region? If so, then why?**

**It is also stated that "[w]e analyse in detail the outputs of the numerical simulations to quantify the role of the different forcing factors." Have these roles not been quantified previously? This**

reviewer is aware of several studies that quantified the relative roles of atmospheric pressures, winds, and waves on the magnitude of storm surge. Do we expect this 1000-storm study to provide new insights? If so, then why?

A — We have modified this paragraph in the introduction to highlight the objectives and the main novelties of this study. First, we perform a basin-scale study, in contrast to earlier assessments that are limited to local cases. Second, we tackle both coastal sea levels and wind-waves, using a coupled model, which is also a new perspective. Third, we investigate the main drivers of coastal hazards, by separating the different sources (pressure, winds and waves). Finally, we exploit a new, large and comprehensive synthetic dataset over the Caribbean. We have referenced the current literature of which we are aware, but we will be keen to cite any other work that the reviewer provides in relation to the topic and that might be missing.)

**(2) — Lines 49–63: Can the authors provide more justification for how the storms were selected? This reviewer is interested in two aspects. First, why use the maximum wind speed (presumably, see minor comment below) as a proxy for the storm intensity? Why not use the minimum central pressure, or maximum radius, or an integrated quantity like the total kinetic energy? The STORM database includes several parameters for each storm, not just the maximum wind speed, but it seems like the current methodology is ignoring them.**

**Second, the random selection of subsamples seems suboptimal. Why not use a maximum dissimilarity algorithm to identify the top 1000 storms that best span the parameter space? Or surely there are other methods that could be considered? The authors' method appears to work okay, given the convergent errors shown in Figure 1, but it would be nice to see a brief discussion of why this method was selected over other options.**

A — To select the subset of TCs from the entire dataset we used maximum wind speed because this is the variable that is better represented in STORM. It shows the highest correlation between the North Atlantic basin in the STORM dataset compared to the IBTrACS dataset [Bloemendaal et al., 2020].

Regarding the methodology, there are certainly other algorithms that serve the same purpose. Our method is simply one of them and we believe is does work properly. We demonstrate so when we compare the distributions of the total dataset and the subsets. We determine objectively the required size of the subsample, and we show that the distributions of the entire dataset and the subsample are consistent temporally and spatially. To put it into context, we have now cited other methods, too.

**(3) — Lines 79–81 and 170–173: Not sure what this means. There were 46 sea-level records in the region – they must have observed something useful about the water levels. The authors refer to a lack of "footprint" – does this mean that the observation records do not include the effects of the storms? The lack of water-level validation is a critical flaw in this study. Somehow the water-level predictions need to be validated, either for these or other storms. Without a validation of the water levels, the rest of the results cannot be trusted.**

A — We agree. This was a weak point also raised by other reviewer, who even provided an additional reference, so we have extended our validation to include measurements from both waves and coastal tide gauges. Please, refer to the new section 3.2.

**(4) — Lines 92–94: References are needed for these methods. Why use a distance of 300 km? Why use a reduction of 20 percent?**

A — We have included the corresponding references. With respect to the 300 km distance, this is justified because the transfer of energy between the atmosphere and the ocean at such a long distance from the TC eye is negligible, according to Holland [1980] and Holland et al. [2010]. As for the 20 % parameter, we have included a reference in the paper [Willoughby and Black, 1996]. We would like to note here that this reduction will not alter the results as we do not account for winds over land.

**(5) — Sections 2.1 and 2.2: There is a mismatch between the atmospheric forcing used for the validation and the rest of the study. The validation storms use a data-assimilated product, which should be accurate (although this reviewer is not convinced that ERA5 can resolve the full dynamics of a tropical cyclone), whereas the synthetic storms are developed with a parametric model. The validation storms have different resolution (1 hr, 30 km) than the synthetic storms (3 hr, 5 km). Can the authors comment on how these differences may affect the validation?**

More importantly, why not generate the atmospheric forcing for Wilma and Tomas in the same way as the synthetic storms? The authors could use the historical track information for these two storms, push it through the parametric model, and then be able to compare apples-to-apples.

A — We agree with the reviewer. We have therefore performed tests and we have changed all the format of validation section in order to add synthetic fields and also more events (related to next question).

(6) — Figure 4: Why use only two buoys per storm? Why not do a comprehensive validation by using all available observations? As-is, the reader can assume that these buoys were cherry-picked to show the best results.

A — It was never our intention to show biased results. The validation of waves with buoys is generally very good, as the winds far from the TC eye are well reproduced. In the new validation section 3.2 we have now included all available data for buoys and tide gauges.

(7) — Lines 184–185: "but the generated waves are less intense due to the smaller fetch area." The Caribbean Sea is a large basin, with a minimum width of 600 km at its narrowest. Why would any waves be fetch-limited in this basin?

A — We meant smaller fetch in comparison to the Atlantic counterpart, where TC have much more space to develop. This has been clarified in the text.

(8) — Lines 204–205 and 294–295: This claim should be explored, ideally via a more-comprehensive validation with the full set of available observations. But more importantly, why do the authors think the wave set-up is under-estimated? Should it be more than 5 percent of the total contribution?

A — For wave setup effects to be well modelled in the nearshore, a high spatial resolution is required. The separation between nearby grid points needs to be able to represent the nearshore slope and the wave breaking. This is not the case with a coastal resolution of $\sim 2$ km. Therefore, it is not possible to validate wave setup when it is not well reproduced by the model. We are aware that it is underestimated in most cases. The work by Amores et al. [2020] demonstrated that, under conditions of strong sustained storms, wave setup may contribute up to 40-50% to total coastal water levels. We believe that, in regions with wide continental shelves and under strong TC, at least part of the wave setup is captured by the model.

(9) — Lines 305–306: The repository should include more than just the return periods. The selected 1000 synthetic storms should be included, both in their parameters from the STORM dataset and their pressure/wind fields from the parametric model. The SCHISM and WWM-III input files should also be included. This will allow for reproducibility of the study results.

A — We initially decided to limit the product to what we consider the most useful (in terms of impact analysis) product of the work. But we agree with the reviewer that reproducibility is very important, so we are now sharing all the outputs in the same data repository 10.5281/zenodo.7069110

(10) — Section 4: An overarching comment is that it is not clear (at least to this reviewer) what is novel about the study findings. It should be expected that the windward islands are affected by Atlantic storms, whereas the west side of the Caribbean is affected by storms from that basin. It should also be expected that regions with narrow shelfs and deep offshore bathymetry will have smaller storm surges that are forced mainly by the storm's pressure deficit, whereas regions with wide shelfs and shallow offshore bathymetry will have larger storm surges that are forced mainly by the storm's winds. Can the authors do more to contextualize their findings and motivate their novelty?

A — Please, see our response above to the main comment (1). We have included some additional text in the introduction to state the major novelties and strengths of the study. We are well aware and agree with the generalities described by the reviewer, but our work quantifies the coastal hazards from storm surges and wind-waves along all the coastlines in the Caribbean induced by TCs and this is a result that, to our knowledge, has not been published before.

Other specific comments:

**(1) — Line 57: "trough" should be 'through' for correct spelling.**

    A — Done

**(2) — Figure 1: It would be helpful to describe what is meant by "speed." This reviewer assumes it is the maximum wind speed at any location/time during the storm. But it could be something else, e.g. the forward speed of the storm.**

    A — The reviewer is correct, is Maximum wind speed and we have changed this in the Figure.

**(3) — Line 65: Again, it is assumed that these speeds (e.g. 111 km/h) refer to the maximum wind speeds, but this should be clarified in the text.**

    A — Done

**(4) — Figure 2: This reviewer struggled to see the tracks and labels in this figure, as they were depicted in black on a mostly blue background. Not sure what to suggest to make these features to be more legible. What if the track and labels were in white?**

    A — This figure has undergone some modifications due to changes in the validation

**(5) — Line 78: Should be 'data were' for subject-verb agreement.**

    A — Changed

**(6) — Line 97: "last" should likely be 'latest.' Please give the actual version numbers for SCHISM and WWM-III.**

    A — Done.

**(7) — Line 122: "where" should be 'were' for correct spelling.**

    A — Done

**(8) — Line 275: "are" should be 'area' for correct spelling.**

    A — Done

**(9) — Line 280: When the letter 'm' is shown in italic font, this reviewer assumes it is a variable, e.g. 25 times m. If it is meant to be a unit (meters), then it should not be in italic font**

    A — Changed

**References**

Nadia Bloemendaal, Ivan D Haigh, Hans de Moel, Sanne Muis, Reindert J Haarsma, and Jeroen CJH Aerts. Generation of a global synthetic tropical cyclone hazard dataset using storm. *Scientific data*, 7(1):1–12, 2020. doi: 10.1038/s41597-020-0381-2.

Greg J Holland. An analytic model of the wind and pressure profiles in hurricanes. *Monthly weather review*, 108(8):1212–1218, 1980.

Greg J Holland, James I Belanger, and Angela Fritz. A revised model for radial profiles of hurricane winds. *Monthly weather review*, 138(12):4393–4401, 2010. doi: https://doi.org/10.1175/2010MWR3317.1.

HE Willoughby and PG Black. Hurricane andrew in florida: Dynamics of a disaster. *Bulletin of the American Meteorological Society*, 77(3):543–550, 1996. doi: https://doi.org/10.1175/1520-0477(1996)077⟨0543: HAIFDO⟩2.0.CO;2.

A. Amores, M. Marcos, D. S. Carrió, and L. Gómez-Pujol. Coastal impacts of storm gloria (january 2020) over the north-western mediterranean. *Natural Hazards and Earth System Sciences*, 20(7):1955–1968, 2020. doi: 10.5194/nhess-20-1955-2020.

---

## Author Comment (AC3)

Response to Reviewer #3 of our manuscript entitled

**Coastal extreme sea levels in the Caribbean Sea induced by tropical cyclones submitted to *Natural Hazards and Earth System Sciences.**

Ariadna Martín, Angel Amores, Alejandro Orfila, Tim Toomey, Marta Marcos

October 7, 2022

This study investigates the storm surge and wind-wave components of extreme sea levels in the Caribbean Sea induced by tropical cyclones. The approach applied in this study is different from previous studies as it is based on a large set of synthetic tracks of tropical cyclones while also taking wind-waves into account. Previous large-scale studies generally excluded wind-waves because of the high model resolution that is required to model the wind-wave component. The study finds that tropical cyclones in the Caribbean come from two well-differentiated families with very distinct intensities and genesis locations. Also, the contribution of each of the forcing mechanisms to the total water level has been investigated. Finally, return levels of wind-waves and sea-surface are provided in a dataset.

Overall I find the study scientifically relevant. However, in my opinion still some substantial improvements are required before this manuscript can be considered for publication in NHESS. My main points of concern are addressed under specific comments. In addition, there are quite some spelling errors. I would advise to let a native English speaker check the manuscript. Some of the writing mistakes that I found are listed under technical corrections, as well as some suggestions to improve on the clarity of the text and figures.

Main comments:

(1) — I am not convinced that the selected subset of 1,000 TC tracks represents the complete 10,000 years of TC activity from STORM. You are arguing that figures 1e, and 1f look very similar. However, I disagree as the patterns don't match. In addition, in the discussion you mention that SSE is very dependent on the morphology of the coastlines. This tells me that even just a very minor shift in storm track could potentially result in a completely different storm surge. The way you checked whether your set of 1,000 TC events represents the STORM dataset (complete 10,000 years) doesn't take this into account.

> A — The method to select the subset of TCs follows Toomey et al. [2022]. Our results in Figure 1a and b in the manuscript indicate that the distribution of maximum velocities from TCs is consistent among both datasets, with correlations over 0.9. We recognise that the qualitative comparison of the maps in Figure 1 e and f may be misleading. However, the statistics are also computed and plotted in Figure 1 c and d. Again, we observe that the correlation among the spatial distribution in the maps is about 0.96, thus confirming their consistency.
>
> In terms of the differences in the results of the two subsets along the coastlines, we refer to the comparison between Figures 3c and S1a of the manuscript, where we have mapped the average number of TC per decade hitting every coastal grid point for the subsample and the entire dataset, respectively. The two maps are indistinguishable.

(2) — Linked to this, the historical dataset IBTrACS that contains observed tracks of TCs shows that the north coast of South America experienced basically zero TCs in the past 40 years. In the STORM dataset there are multiple, even within just a decade of data. Most likely due to the way the STORM model was set up, which leaves some freedom to the TCs to travel in a certain

random direction, next to the most common north-west direction for the Atlantic basin. How is the uncertainty related to this represented and illustrated in your results?

A — We are aware of this issue and is an inherent problem with the original data base. Indeed, this is related to the large standard deviation that IBtrACS has in the number of landfalls, that indicates a substantial difference in the year-to-year landfall counts. This is reflected in the STORM dataset. However, on average, the landfall counts for both datasets are within one standard deviation of each other [Bloemendaal et al., 2020]. We have added a comment in the paper to indicate it.

(3) — The different settings that you used for the hydrodynamic model are described in the methods section. However, I am missing an explanation why you chose those settings. For example, why did you use the Pond & Pickard formulation to calculate the wind stress?

A — All our settings are based on both previous studies and tests conducted during the validation process. In particular Pond & Pickard formulation is only used in the section of the contributions to SSE, while for the fully coupled run the wind stress is calculated directly using the forcing fields which proved to be superior to the former when waves are available [Bertin et al., 2015]. This is added in the text.

(4) — Hurricanes Wilma and Thomas are used to validate the numerical simulation set-up. However, no observations are available from nearby tide gauges at the time of these two tropical cyclones. Does this mean that validation of the hydrodynamic model is completely lacking? I don't understand why you pick these events if observations from tide gauges are unavailable? Would it be possible to simulate some other tropical cyclone events for which tide gauge observations are available?

A — In the revised version we have extended the validation to include tide gauges and more buoys. Now, section 3.2 presents a complete validation of the model setup using all in-situ measurements available.

(5) — Suggestion: did you compare SSE return periods, so excluding waves, with the COAST-RP dataset from Dullaart et al. (2021)? It would be interesting to compare because the input dataset, being STORM, is the same. However, this study used 3,000 years of TC activity from the STORM dataset instead of 1,000 events like you did. I realize that COAST-RP includes tidal levels as well, but because the tidal range is very small in this area a comparison could be of added value.

A — Thank you for the suggestion. We have compared both results and the conclusions are cited the paper in our discussion.

(6) — The relatively coarse coastal resolution of the model grid results in an underestimation of the wave set-up (line 204). This is a major limitation of this study correct? Then why isn't it discussed later on in the discussion section?

A — We explain that our information provides the areas of greatest impact against the SSE, where we provide a threshold value because we may well be underestimating the contribution of the wave setup. In Sec. 4 (Lines: 314-315) we underline these limitations and explicitly discuss this point. To increase the coastal resolution to achieve better results is beyond of the scope of the present manuscript. We are doing this computation in the Mediterranean Sea and requires a lot of computational effort.

(7) — At this point, I am not convinced that performing the study again will result in the same findings. I believe this is crucial for all scientific studies. Improving the clarity of the methods section could be the first step here.

A — We have modified the text according to the suggestions provided by the reviewer above. We hope that these changes will help clarifying the results. In addition, and following the request from reviewer 2, we have now shared all the outputs of the study, for the sake of reproducibility. New repository: 10.5281/zenodo.7069110

Other specific comments:

**(1) — You refer to the hydrodynamic model as 'hydrodynamic' or 'numerical' model. Please be consistent.**

A — Changed

**(2) — Line 12: "Here we focus in" -¿ "here we focus on**

A — Changed

**(3) — Line 13: "TC" -¿ "TCs". Please check throughout the manuscript.**

A — Done

**(4) — Line 20: "small islands" -¿ "small island"**

A — Changed

**(5) — Line 23: "GPD" -¿ "GDP"**

A — Done

**(6) — Line 58: What do you mean by maximum speed distribution?**

A — Maximum wind speed distribution. Is already changed in the text.

**(7) — Line 85: I believe this paragraph could be improved. It might help the reader to see a figure of the wind field generated using the holland model. In addition the last sentence requires some explanation. Why do you reduce the velocity by 20% over land areas? Do you maybe have a reference for that?**

A — We have used a state of the art approach here. Examples of the Holland wind profiles can be found easily, so we do not feel it is needed to be included in the manuscript. If we have misunderstood the comment, we would be keen to add more information in this respect. We added a reference on the 20% velocity reduction over land [Willoughby and Black, 1996]. We would like to remark that this reduction will not affect the results at all, since we do not account for impacts of wind over land.

**(8) — Line 110: How do you know that the selected domain is large enough to allow for a correct generation and propagation of the wind-waves originated by hurricanes affecting the Antilles? Did you perform a test run for this?**

A — We conducted a test in which we represent the values of $H_s$ generated by a TC at a range of distances from a point where in-situ buoy measurements are available. The distances vary between 0 and 8000 km. An example is illustrated in Figure 1 (number of figure refers to this document) for one of the buoys. The test has been carried out for every available buoy within or close to our survey domain and for all TC in the IBTrACS dataset. In the figure it is observed that the values of $H_s$ decrease rapidly between 0 and 1000 km. On average, when all buoys are considered, $H_s$ lies below 1 m between 1000 and 2000 km distance. We therefore conclude that 2000 km is a distance long enough for the forcing of the TC to develop the wave field.

**(9) — Line 203: This sentence seems incomplete.**

A — We have rewritten the sentence

**(10) — Line 219: Shouldn't this be part of the methods section?**

A — This part with the description of the methodology has been moved to the methods section (new subsection 2.4).

**(11)** — **Line 251: In this paragraph you are describing some storm characteristics. I don't believe this belongs in the summary and discussion section. Instead, maybe put it under the introduction?**

A — We understand this point of view. However, we compare our results with past events and we believe that it is important to keep all the information together, so we prefer to leave it in this section.

**(12)** — **Line 288: duplicate of "to"**

A — Changed

**(13)** — **Figure 1: This figure is very hard to digest. The letters indicating the subpanels are sometimes hard to see due to the dark background colours. I would suggest you put them just outside of the panel. For consistency it would be nice if you do the same with the other figures. Also, it would be good to reduce the number of subpanel titles and make sure that they are in the same location each time. So for example, in the top left. Last, the figure caption includes f) and g) which should be e) and f). I believe c) is missing.**

A — We have added white background to the text of the lower panels to facilitate reading and we also changed the typos in the caption. We have preferred to keep the letters and text inside the plots, as it allows to maintain a larger size in the panels.

**(14)** — **Figure 2: black lines and text on a dark blue background is not a great match.**

A — This figure has undergone some modifications, including the removal of the tracks, due to changes in the validation.

**(15)** — **Figure 3: panel b shows a percentage correct? So a positive 100 % means that for every tropical cyclone with Caribbean origin, there are 0.5 cyclones with an Atlantic origin? If so, the percentage will exceed 100% in some locations correct? Right now the maximum value is 100% according to the colour bar. In addition, you mention "radius of maximum speed". Do you mean radius of maximum winds? Rmax is more commonly used as an abbreviation for this.**

A — Former panel b (now panel d) shows the percentage of the dominant family (of Atlantic or Caribbean origin) at each grid point. We have modified the figure caption to make it clearer. What we show here is that 100% (red) means that all TC affecting that point are of Caribbean origin, while -100% (blue) indicates that all are of Atlantic origin (the eastern Antilles being a good example).
We have changed radius of maximum speed by Rmax, as suggested.

**(16)** — **Figure 5: caption -¿ What do you mean by 'poor shore resolution'?**

A — We meant here that we have a relatively coarse resolution ($\sim 2$ km) to represent nearshore coastal processes and therefore we represent the points that are clsoe to the shoreline but at 20 m depth. The text has been slightly modified.

**References**

Tim Toomey, Angel Amores, Marta Marcos, Alejandro Orfila, and Romualdo Romero. Coastal hazards of tropical-like cyclones over the mediterranean sea. *Journal of Geophysical Research: Oceans*, 127(2): e2021JC017964, 2022. doi: https://doi.org/10.1029/2021JC017964.

Nadia Bloemendaal, Ivan D Haigh, Hans de Moel, Sanne Muis, Reindert J Haarsma, and Jeroen CJH Aerts. Generation of a global synthetic tropical cyclone hazard dataset using storm. *Scientific data*, 7(1):1–12, 2020. doi: 10.1038/s41597-020-0381-2.

Xavier Bertin, Kai Li, Aron Roland, and Jean-Raymond Bidlot. The contribution of short-waves in storm surges: Two case studies in the bay of biscay. *Continental Shelf Research*, 96:1–15, 2015.

HE Willoughby and PG Black. Hurricane andrew in florida: Dynamics of a disaster. *Bulletin of the American Meteorological Society*, 77(3):543–550, 1996. doi: https://doi.org/10.1175/1520-0477(1996)077⟨0543: HAIFDO⟩2.0.CO;2.

[Figure]

Figure 1: Example for buoy 42001 of the calculation of the $H_s$ as a function of distance from the buoy, using all the TCs in the IBTrACS database that affected that buoy.This particular buoy is located in the Golf of Mexico, near the Yucatan channel.. Each TC is coloured according to its lifetime at each point, and the black line represents the median $H_s$ value of all TCs as a function of distance.

---

## Referee Report (RR1)

Review of the manuscript: "**Coastal extreme sea levels in the Caribbean Sea induced by tropical cyclones**" by Ariadna Martín et al.

This is my second review of the paper. I want to thank the authors' work and effort answering my recommendations to the first version of the manuscript, which I believe helped to improve the paper. The validation section was completely re-written, giving from my point of view a better foundation to the results. I still think the investigation contribution is interesting and scientifically relevant, with useful results for coastal risk assessment associated to tropical cyclones in the Caribbean Sea. However, in my opinion some aspects need to be improved. For this reason, my recommendation is a minor revision before considering its publication in the Natural Hazards and Earth System Sciences journal.

Specific comments:

(1) L57. Review the sentence structure.

(2) L61-63. In my previous review, I asked to clarify how the PDFs were built. Although authors improved the description, I still have problems understanding this procedure, which is important, as it is the base of using a 1000 TCs to represent the complete dataset. I suggest including the maximum wind speed and the spatial distribution of the TCs track PDFs in a two panels figure in the supplement material. In each case, show the PDFs from the complete dataset (25494 TCs) and from 1000 events sub-set, as this was selected as the proper number of TCs to represent the complete dataset. Please see my comment (23) to Figure 1.

(3) L62. Remove "(".

(4) L63. … "for the maximum wind speed" …

(5) L82. "area". "Caribbean Sea".

(6) L96. "five real TCs".

(7) L141-142. "GPD fitted to all measurements". I suggest replacing measurements for "synthetic values" or similar, as return levels are not constructed from observations.

(8) L156-158. I bring again this comment, as I fail to explain myself in the previous review (comment 9). Indeed, the TCs prevailing travelling direction in the Caribbean is toward the west-northwest. As commented by Torres and Tsimplis (2014), "Due to the diminution of the Coriolis force close to the equator, any tropical cyclones passing toward the south of the basin are weak. South of 10°N there is less than 1% chance of a hurricane strike per year [Pielke et al., 2003]". Therefore, my suggestion was to consider if you wanted to include a comment about the relation between the weak Coriolis force toward the south of the Colombian Basin and the smaller number of TCs per decade seen in this region.

(9) L179-181. Please clarify this sentence. You compare landfalls per year, but figure 3 shows landfalls per decade. Besides, where the reader can see the results form IBTrAcs? It is in one of the Knapp et al. papers or is your own calculation, which is not shown?

(10) L184. For the first time in the paper the variable "$H_s$" appears, therefore please indicate its name. This variable is usually used to define "significant wave height". By definition, "significant wave height" indicates the mean wave height of the highest one-third of the waves. Through the paper, $H_s$ is not used following the pervious definition, e.g. Figure 5ab, "a) and b) represent the 99$^{th}$ percentile of the maximum $H_s$ …". Please consider changing this variable (could be "wave height") and review the correct use of the term "significant wave height" in all the manuscript.

(11) L192. The agreement is not so good in SSE. As you mention in line 288, your SSE includes "only the hydrodynamic response" to wind, pressure and waves. Therefore, the lack of a better agreement is probably because the observed SSE includes the tide, while your simulated SSE does not. Although the Caribbean has a microtidal environment, this can be important for extreme SSE. I recommend that for the comparison shown in Fig 4b, you use the tidal residual from the observed sea level time series. This can also have an important effect when fulfilling the 0.4 m peak criteria used to detect cyclone-related SSE in sea level time series (L190).

(12) L210. Please verify the referenced figure.

(13) L222-225. I found more interesting the SSE results when they are shown as relative terms (Fig.S2 - contribution percentage), when compared to absolute values as shown in Fig.6. This is because in my view, the paper provides a statistical perspective of the TC effects in the Caribbean Sea, what I found more important than the absolute values presented. Please consider to switch these two figures.

(14) L230. I do not see how figure S3 supports the relationship between the distance to the eye and the atmospheric pressure contribution to SSE. Please see comments to Figure S3 (28).

(15) L237. "the model's spatial resolution".

(16) L243. Consider replacing "northern coast" by "northern Caribbean boundary", or similar. This because the northern coast can be understood as the Atlantic coast of the Greater Antilles.

(17) L248. In my previous review, I made a suggestion to re-arrange the Summary and discussion section. The authors' response provided an outline of this section with a good structure. However, in my view, a problem remains. In the paragraph that starts in L263, you start discussing the wave results, but from line 265 to 275, you give some examples of historical major impacts from hurricanes in the region. However, not all these impacts are limited to the wave height effect, but probably

including storm surges, and others effects. Therefore, I suggest moving this section before of the last paragraph, which starts in L314.

(18) L253. It is mentioned here for the first time in the paper that the "Caribbean basin family generates off the cost of Honduras". This exact place of generation of the Caribbean TCs family was not discussed previously in the results section. Besides, in my view, this statement is inaccurate, as in Figure 3e, TCs effects from this family are seen even in the Venezuela basin. I mention this issue again, as my comment 22 of the previous review was not clearly answered by the authors.

(19) L261-262. This line was included, answering to my comment (24) from my previous review. I recommend including the reference, so readers can know the origin of this statement.

(20) L279-280. Please consider including a value or range of the wave height 30-year return level found by Montoya et al (2018), as I believe it might serve as a reference to the 100-year return levels found in your research.

(21) L282. "Colombia Basin".

(22) L298. Please verify the referenced figure.

Comments to figures:

(23) Figure 1. Please see my comment (2). ¿Why the vertical axis in panel c) is in meters, if the spatial domain was divided into 2 degrees bins? Below the colorbar there is a title "Normalized Spatial distribution (%hurricanes (TCs)/pixel)". ¿How this distribution was normalized? Besides, in L62 you clarify that the PDF is built using the 3-hourly time step for each TC passing though each pixel of the grid. Therefore I am unsure if panels e-f are showing the "% of TCs/pixel" or the "% of TCs hits/pixel", understanding a "hit" as each time the 3-hourly TC position is placed in the pixel. Based on this, if necessary, update L67.

(24) Figure 2. A buoy used to validate Hurricane Ernesto (black dot) is shown in the northern boundary of the study area, which is probably a mistake. Try to improve the description of the location of buoys and tide gauges in the legend.

(25) Figure 3. Title of panel (a) and in the legend replace "mean" for "median", as stated in L153. Consider keeping the range of the color scale for panels a, e and f between e.g. 50-200 km/h, so an easier comparison between these results can be done by the reader. Same comment applies for the color scale range in Fig. S1c and d. In the third line of the legend replace "if" for "it".

(26) Figure 4. My comment (18) from the previous review was about a probably too long wave period (14-18 s) reported inside the Caribbean Sea. In the answer to that comment, you included a validation of the period comparing the model results and buoy data. I suggest including in Figure 4 the validation of the maximum period between the model and the buoy data as forced by TCs, in a similar way as the wave

height is presented in panel (a). Besides, include a grid and/or line of equal observed and simulated values, to facilitate the results assessment. At the end of the legend I suggest to include "… between the eye of the TC and each instrument at the moment of the largest observed value", or similar clarification.

(27) Figure 6. First line of the legend: "a value that".

(28) Figure S3. (a) Seems to be the same as Fig S1a, but with different color scale values; I do not understand the maximum value of 100 TCs. (b) Seems to be the same Fig S2d; the title inside indicates meters what is not coherent with the color scale legend (%). (c) I am not sure that I understand this figure. ¿It tries to support that the larger the number of TCs affecting a node, the larger the atmospheric pressure contribution to SSE?  Please see my comment (14).

---

## Referee Report (RR2)

**REVIEW, "COASTAL EXTREME SEA LEVELS ..."**

MARTÍN, AMORES, ORFILA, TOOMEY, MARCOS

In their manuscript "Coastal extreme sea levels in the Caribbean Sea induced by tropical cyclones," the authors Martín *et al.* use a new database of synthetic tropical cyclones as forcing to simulations of wind waves and storm surges in the Caribbean Sea. It is shown that the wind waves and storm surges vary significantly at coasts around the basin, due to differences in storm evolution, local bathymetry, and other characteristics.

It is my recommendation that this manuscript should be **revised**.

The authors should be commended for revising the Validation to use a consistent atmospheric forcing and to include comparisons to as many observations as possible. That said, the validation for the water levels is unconvincing, with some large errors between observation and model. The authors have selected a threshold water level of 0.4 m to identify observations to include in the validation. This value is somewhat arbitrary? Can the set of available observations be expanded if this threshold is relaxed?

This reviewer also wants to push again on the novelty of the study. The authors have done a better job of emphasizing the lack of comprehensive studies in this region with large numbers of storms and a large geographic coverage, and thus this study does fill a gap in terms of available data. But what does it add to our scientific understanding of storm-induced hazards in the region? As-is, the Discussion confirms findings from observations and other studies. The largest waves affect the Lesser Antilles, West Indies, and northern Caribbean ... which is known from historical storms Hugo, Maria, Irma, and David and recent studies by Pillet and Montoya. The largest water levels are found in Cuba, Mexico, and Belize ... which matches the findings by Torres and Dullaart. The atmospheric pressure has its largest effect along the storm track, whereas the wind forcing has its largest effect in shallow coastal areas ... again, this is known. Is it possible for the authors to extract more understanding from this great new database?

The following major comments can be considered in a revised manuscript:

– For the Validation, it is not clear if tides are included. This reviewer guesses not – can this be clarified?

– Relatedly, for both the Validation and Results, if the tides are excluded, then maybe 'sea surface elevation' is not the best term. It would be better to refer to 'storm surge' or 'non-tidal residual.'

– Lines 75–84 and Table 1: For the water levels, it is mentioned that the gauges were selected if they had observed peaks larger than 0.4 m. For the wind-waves, how were the buoys identified in Table 1 – was there a similar threshold for the peak in significant wave height?

– Lines 93–94: There are newer ways to reduce momentum transfer in overland regions based on land-use/land-cover data. The method in this study (with a uniform 20 percent reduction) is likely okay because the computational domain does not contain a significant amount of overland regions, and the analyses do not focus on them. Can these points be noted here?

– Lines 117–118: Can the authors provide a reference to support this statement?

The following minor comments can also be considered:

– Lines 16 and 156: "over" is a spatial relation, better to use 'more than' here.

– Line 20: 'nations' should be plural.

– Lines 26 and 186: "since" is a temporal relation, better to use 'because' here.

– Line 82: "are" should be 'area'.

– Line 84: 'Figure' should be capitalized.

– Line 97: "lastest" is misspelled.

– Lines 101–102 and 106–107, and page 9 footnote: Can the URLs be moved into the list of references?

– Line 109: Here, "Fig." is abbreviated, but on the preceding page, "Figure" is spelled fully. Please be consistent. See also line 152, etc.

– Line 122: "In order" can be deleted.

– Line 163: "Tcs" is mis-capitalized.

– Line 186: "fullfil' is misspelled.

– Line 187: "Figure Fig." is redundant.

– Line 220: When the letter 'm' is shown in italic font, this reviewer assumes it is a variable, e.g. 25 times m. If it is meant to be a unit (meters), then it should not be in italic font.

– Line 239: Add spaces between the years in this list.

– Line 250: "a" can be deleted.

– Line 268: "hurricane" should be capitalized.

– Lines 299–300: Can this sentence be rewritten for clarity?

– Line 300: "In fact" can be deleted.

– Figure 1 caption: In the second sentence, starting with "Where" is awkward – should this instead be a continuation of the first sentence? In the third sentence, the word "represents" should not be plural.

– Figure 3: For panels (a) and (b), why not use 'intensity' or 'maximum wind speeds' as labels for both plots?

– Figure 3 caption: "when if is within" should be corrected.

– Figure 6 caption: "taht" is misspelled.

– Figure 7 caption: "Levels" should not be capitalized.

---

## Author Response (AR3)

Response to Reviewer #1 of our manuscript entitled

**Coastal extreme sea levels in the Caribbean Sea induced by tropical cyclones submitted to *Natural Hazards and Earth System Sciences.**

Ariadna Martín, Angel Amores, Alejandro Orfila, Tim Toomey, Marta Marcos

December 20, 2022

This is my second review of the paper. I want to thank the authors' work and effort answering my recommendations to the first version of the manuscript, which I believe helped to improve the paper. The validation section was completely re-written, giving from my point of view a better foundation to the results. I still think the investigation contribution is interesting and scientifically relevant, with useful results for coastal risk assessment associated to tropical cyclones in the Caribbean Sea. However, in my opinion some aspects need to be improved. For this reason, my recommendation is a minor revision before considering its publication in the Natural Hazards and Earth System Sciences journal

**Specific comments:**

**(1) — L57.** Review the sentence structure.

> **A —** The sentence has been rewritten

**(2) — L61-63.** In my previous review, I asked to clarify how the PDFs were built. Although authors improved the description, I still have problems understanding this procedure, which is important, as it is the base of using a 1000 TCs to represent the complete dataset. I suggest including the maximum wind speed and the spatial distribution of the TCs track PDFs in a two panels figure in the supplement material. In each case, show the PDFs from the complete dataset (25494 TCs) and from 1000 events sub-set, as this was selected as the proper number of TCs to represent the complete dataset. Please see my comment (23) to Figure 1.

> **A —** We explain the methodology in the manuscript and refer to ? for more information. We agree that the information requested by the reviewer is necessary but, to our understanding, it is the same that is included in Fig.3 and Fig.S1, where we show the wind speed distribution for the subset and the complete dataset respectively. Panels e) and f) of Fig.1 shows the same for the spatial distribution. Thus, we do not see the need to repeat the information.

**(3) — L62.** Remove "(".

> **A —** Done

**(4) — L63.** ... "for the maximum wind speed" ...

> **A —** Done

**(5) — L82.** "area". "Caribbean Sea".

> **A —** Done

(6) — L96. "five real TCs".

    **A** — Done

(7) — L141-142. "GPD fitted to all measurements". I suggest replacing measurements for "synthetic values" or similar, as return levels are not constructed from observations.

    **A** — Done

(8) — L156-158. I bring again this comment, as I fail to explain myself in the previous review (comment 9). Indeed, the TCs prevailing travelling direction in the Caribbean is toward the west-northwest. As commented by Torres and Tsimplis (2014), "Due to the diminution of the Coriolis force close to the equator, any tropical cyclones passing toward the south of the basin are weak. South of 10°N there is less than 1% chance of a hurricane strike per year [Pielke et al., 2003]". Therefore, my suggestion was to consider if you wanted to include a comment about the relation between the weak Coriolis force toward the south of the Colombian Basin and the smaller number of TCs per decade seen in this region.

    **A** — We thank the reviewer for this suggestion. We now understand the point and have included the text accordingly in the manuscript, along with the reference provided

(9) — L179-181. Please clarify this sentence. You compare landfalls per year, but figure 3 shows landfalls per decade. Besides, where the reader can see the results form IBTrAcs? It is in one of the Knapp et al. papers or is your own calculation, which is not shown?

    **A** — The comparison between STORM and IBTrAcs is extracted from ?. Here we just emphasise some important aspects of the study that should be taken into account. For clarification, we have added the reference in the manuscript.

(10) — L184. For the first time in the paper the variable "Hs" appears, therefore please indicate its name. This variable is usually used to define "significant wave height". By definition, "significant wave height" indicates the mean wave height of the highest one-third of the waves. Through the paper, Hs is not used following the pervious definition, e.g. Figure 5ab, "a) and b) represent the 99th percentile of the maximum Hs ...". Please consider changing this variable (could be "wave height") and review the correct use of the term "significant wave height" in all the manuscript

    **A** — We have added the symbols $H_s$, $T_p$ and $D_p$ the first time their name is mentioned (L 119-120). Here we use the same definition of $H_s$: mean wave height of the highest one-third of the waves, as it is the variable that provides the model. When referencing e.g to the 99th percentile of the maximum of $H_s$, one has to understand that the model provides $H_s$ for each time step for each TC for all the grid points. To represent that on the map, first we use the maximum value of $H_s$ affecting each coastal point by each TC, obtaining (# of coastal points) x (# TCs, in our case 1000) values. Then, we compute the $99_{th}$ percentile of that values to obtain a single value for each coastal point.

(11) — L192. The agreement is not so good in SSE. As you mention in line 288, your SSE includes "only the hydrodynamic response" to wind, pressure and waves. Therefore, the lack of a better agreement is probably because the observed SSE includes the tide, while your simulated SSE does not. Although the Caribbean has a microtidal environment, this can be important for extreme SSE. I recommend that for the comparison shown in Fig 4b, you use the tidal residual from the observed sea level time series. This can also have an important effect when fulfilling the 0.4 m peak criteria used to detect cyclone-related SSE in sea level time series (L190).

    **A** — We apologise for the missing information. Indeed, for the comparison between observed and modelled SSE, we have used the non-tidal residual of the tide gauges. Prior to the validation we removed the tides using UTide function. We have added this information to the methodology section (L 80-82)

(12) — L210. Please verify the referenced figure.

  **A — It is correct**

(13) — **L222-225.** I found more interesting the SSE results when they are shown as relative terms (Fig.S2 - contribution percentage), when compared to absolute values as shown in Fig.6. This is because in my view, the paper provides a statistical perspective of the TC effects in the Caribbean Sea, what I found more important than the absolute values presented. Please consider to switch these two figures.

  **A — We are grateful for the suggestion. We have considered switching Fig 6 and Fig S2. However, we believe that this would be inconsistent with the rest of the results shown in the manuscript, which are given in absolute terms. This is why we prefer to keep the figures as they are now. Nevertheless, we agree the information is relevant, but we hope that the interested reader can access easily to the supplementary material when needed.**

(14) — **L230.** I do not see how figure S3 supports the relationship between the distance to the eye and the atmospheric pressure contribution to SSE. Please see comments to Figure S3 (28).

  **A — The reviewer is right. In figure S3 we are only illustrating the relationship between the distance of the TC to the coastline and pressure effect. The text has been modified accordingly.**

(15) — **L237.** "the model's spatial resolution".

  **A — Done**

(16) — **L243.** Consider replacing "northern coast" by "northern Caribbean boundary", or similar. This because the northern coast can be understood as the Atlantic coast of the Greater Antilles.

  **A — Done**

(17) — **L248.** In my previous review, I made a suggestion to re-arrange the Summary and discussion section. The authors' response provided an outline of this section with a good structure. However, in my view, a problem remains. In the paragraph that starts in L263, you start discussing the wave results, but from line **265 to 275**, you give some examples of historical major impacts from hurricanes in the region. However, not all these impacts are limited to the wave height effect, but probably including storm surges, and others effects. Therefore, I suggest moving this section before of the last paragraph, which starts in L314.

  **A — The have now rearranged the summary. Following reviewer's suggestion we have moved the references to the impact of TC in the Caribbean. However, instead to moving this part to the end of the summary, we have used it as a starting point. As it is now the references are not inconsistent in terms of the causes of the TC impacts (not necessarily waves only).**

(18) — **L253.** It is mentioned here for the first time in the paper that the "Caribbean basin family generates off the cost of Honduras". This exact place of generation of the Caribbean TCs family was not discussed previously in the results section. Besides, in my view, this statement is inaccurate, as in Figure 3e, TCs effects from this family are seen even in the Venezuela basin. I mention this issue again, as my comment **22** of the previous review was not clearly answered by the authors.

  **A — There might be a misunderstanding here. We refer to the formation area, not to the impacting area in this sentence, which can indeed affect other regions such as the Venezuela basin.**

(19) — **L261-262.** This line was included, answering to my comment (24) from my previous review. I recommend including the reference, so readers can know the origin of this statement.

  **A — Done**

**(20)** — **L279-280. Please consider including a value or range of the wave height 30-year return level found by Montoya et al (2018), as I believe it might serve as a reference to the 100-year return levels found in your research.**

    **A** — Done

**(21)** — **L282. "Colombia Basin".**

    **A** — Done

**(22)** — **L298. Please verify the referenced figure.**

    **A** — Done

Comments to figures:

**(23)** — **(23) Figure 1. Please see my comment (2). ¿Why the vertical axis in panel c) is in meters, if the spatial domain was divided into 2 degrees bins? Below the colorbar there is a title "Normalized Spatial distribution (%hurricanes (TCs)/pixel)". ¿How this distribution was normalized? Besides, in L62 you clarify that the PDF is built using the 3-hourly time step for each TC passing though each pixel of the grid. Therefore I am unsure if panels e-f are showing the "% of TCs/pixel" or the "% of TCs hits/pixel", understanding a "hit" as each time the 3-hourly TC position is placed in the pixel. Based on this, if necessary, update L67.**

    **A** — The reviewer is right, the units in panel c) are wrong. This, along with a clarification on how the distribution was normalized has been added to the legend. With regards to how the PDF was built is express on L63: "counting each time step".

**(24)** — **Figure 2. A buoy used to validate Hurricane Ernesto (black dot) is shown in the northern boundary of the study area, which is probably a mistake. Try to improve the description of the location of buoys and tide gauges in the legend.**

    **A** — There is no mistake in the location of the buoy. The fact that it seems to be outside the domain is due to the size of the dot, which is chosen to be wider than the boundary line. We have also modified the figure caption to improve the description.

**(25)** — **Figure 3. Title of panel (a) and in the legend replace "mean" for "median", as stated in L153. Consider keeping the range of the color scale for panels a, e and f between e.g. 50-200 km/h, so an easier comparison between these results can be done by the reader. Same comment applies for the color scale range in Fig. S1c and d. In the third line of the legend replace "if" for "it".**

    **A** — We have corrected the title. We have not made any changes to the colour scales because the geographical patterns are hardly visible if we expand them as suggested by the reviewer.

**(26)** — **Figure 4. My comment (18) from the previous review was about a probably too long wave period (14-18 s) reported inside the Caribbean Sea. In the answer to that comment, you included a validation of the period comparing the model results and buoy data. I suggest including in Figure 4 the validation of the maximum period between the model and the buoy data as forced by TCs, in a similar way as the wave height is presented in panel (a). Besides, include a grid and/or line of equal observed and simulated values, to facilitate the results assessment. At the end of the legend I suggest to include ". . . between the eye of the TC and each instrument at the moment of the largest observed value", or similar clarification.**

    **A** — We have added a panel to the validation figure (Fig.4) to show the wave period validation for all the buoys used for the $H_s$. We have clarified the legend accordingly. Regarding the grid in the figure, we feel that it hampers the comparison rather than facilitating it, as it makes the plot more blurry.

**(27)** — **Figure 6. First line of the legend: "a value that".**

**A** — **Done**

**(28)** — **Figure S3. (a) Seems to be the same as Fig S1a, but with different color scale values; I do not understand the maximum value of 100 TCs. (b) Seems to be the same Fig S2d; the title inside indicates meters what is not coherent with the color scale legend (%). (c) I am not sure that I understand this figure. ¿It tries to support that the larger the number of TCs affecting a node, the larger the atmospheric pressure contribution to SSE? Please see my comment (14).**

**A** — **The name in panel b) has been corrected. As for the rest of the comments, yes panel a) and b) are shown in other figures. Please, refer to our response to comment (14) for Fig. S3.**

Response to Reviewer #2 of our manuscript entitled

**Coastal extreme sea levels in the Caribbean Sea induced by tropical cyclones submitted to *Natural Hazards and Earth System Sciences.**

Ariadna Martín, Angel Amores, Alejandro Orfila, Tim Toomey, Marta Marcos

December 20, 2022

In their manuscript "Coastal extreme sea levels in the Caribbean Sea induced by tropical cyclones," the authors Martín et al. use a new database of synthetic tropical cyclones as forcing to simulations of wind waves and storm surges in the Caribbean Sea. It is shown that the wind waves and storm surges vary significantly at coasts around the basin, due to differences in storm evolution, local bathymetry, and other characteristics.

**Main comment:**

**(1)** — The authors should be commended for revising the Validation to use a consistent atmospheric forcing and to include comparisons to as many observations as possible. That said, the validation for the water levels is unconvincing, with some large errors between observation and model. The authors have selected a threshold water level of 0.4 m to identify observations to include in the validation. This value is somewhat arbitrary? Can the set of available observations be expanded if this threshold is relaxed?

> **A** — We agree with the reviewer and we had indeed tested different thresholds. Of course, a lower threshold would result in larger number of values, but then the question is whether those maxima are indeed related to the passage of TC or to any other disturbances present in the observations. The selection of the threshold is therefore a trade-off of the number of peaks and the real signals. Despite the limitations of the results we believe that this threshold is good enough. We have tested it and no improvements were found with other cases..

**(2)** — This reviewer also wants to push again on the novelty of the study. The authors have done a better job of emphasizing the lack of comprehensive studies in this region with large numbers of storms and a large geographic coverage, and thus this study does fill a gap in terms of available data. But what does it add to our scientific understanding of storm-induced hazards in the region? As-is, the Discussion confirms findings from observations and other studies. The largest waves affect the Lesser Antilles, West Indies, and northern Caribbean ... which is known from historical storms Hugo, Maria, Irma, and David and recent studies by Pillet and Montoya. The largest water levels are found in Cuba, Mexico, and Belize ... which matches the findings by Torres and Dullaart. The atmospheric pressure has its largest effect along the storm track, whereas the wind forcing has its largest effect in shallow coastal areas ... again, this is known. Is it possible for the authors to extract more understanding from this great new database?

> **A** — The information we provide in the manuscript may not seem groundbreaking. However, as we emphasise in the introduction (L37-39), the importance of the study lies in the numerical quantification of these results. In addition, having done a study that encompasses the entire Caribbean coastline, our results focus on highlighting those areas most affected and understanding why, while doing an effort to compare these results with historical data. In addition, the results of this work can be used for the study of smaller

or more localised areas where until now there have been insufficient data. The data we provide from the model can be used for coastal protection studies or to extend our results to more specific areas with higher spatial resolution. All the data are freely available.

Other specific comments:

(1) — For the Validation, it is not clear if tides are included. This reviewer guesses not- can this be clarified?

A — Prior to the validation we removed the tides using UTide function. We have added this information to the methodology section (L 80-82)

(2) — Relatedly, for both the Validation and Results, if the tides are excluded, then maybe 'sea surface elevation' is not the best term. It would be better to refer to 'storm surge' or 'non-tidal residual.'

A — The standard term for the position of the water surface is sea surface height (see for example the outputs in Copernicus Marina Data Store. Our model provides the variable elevation, so we have modified the term as sea surface elevation. We agree that when dealing with observations it is more common, and probably more understandable, to use either storm surges or tidal residuals. But we feel that SSE is also suitable for the modelled data in our case and see no reason to change it.

(3) — Lines 75-84 and Table 1: For the water levels, it is mentioned that the gauges were selected if they had observed peaks larger than 0.4 m. For the wind-waves, how were the buoys identified in Table 1 - was there a similar threshold for the peak in significant wave height?

A — In the case of wave validation there was no need to use a threshold because we found numerous buoys that could be used simply by visual inspection, unlike for the sea level time series. We used a threshold for the SSE validation because it was difficult to find an event with available data. So, instead of checking all the periods visually (which would have taken a long time), we did the selection in an automatic way, thus needing a definition of event.

(4) — Lines 93-94: There are newer ways to reduce momentum transfer in overland regions based on land-use/land-cover data. The method in this study (with a uniform 20 percent reduction) is likely okay because the computational domain does not contain a significant amount of overland regions, and the analyses do not focus on them. Can these points be noted here?

A — We agree but, in our case, this is not really a relevant factor, as those winds do not generate waves or storm surges, once the TC has made landfall. We only included it for completeness of the method.

(5) — Lines 117-118: Can the authors provide a reference to support this statement?

A — A reference to Bertin et al. (2015) has been included in the revised version of the manuscript.

(6) — Lines 16 and 156: "over" is a spatial relation, better to use 'more than' here.

A — Done

(7) — Line 20: 'nations' should be plural.

A — Done

(8) — Lines 26 and 186: "since" is a temporal relation, better to use 'because' here.

A — We understand that both are acceptable.

**(9)** — Line 82: "are" should be 'area'.

    **A —** Done

**(10)** — Line 84: 'Figure' should be capitalized.

    **A —** Done

**(11)** — Line 97: "lastest" is misspelled.

    **A —** Corrected

**(12)** — Lines 101-102 and 106-107, and page 9 footnote: Can the URLs be moved into the list of references?

    **A —** We are unsure about the criterion of the journal in this respect. We expect that this will be assessed at the editorial level, if the manuscript is accepted for publication.

**(13)** — Line 109: Here, "Fig." is abbreviated, but on the preceding page, "Figure" is spelled fully. Please be consistent. See also line 152, etc.

    **A —** We have corrected this.

**(14)** — Line 122: "In order" can be deleted.

    **A —** Done

**(15)** — Line 163: "Tcs" is mis-capitalized.

    **A —** Corrected

**(16)** — Line 186: "fullfil" is misspelled.

    **A —** Corrected

**(17)** — Line 187: "Figure Fig." is redundant.

    **A —** Corrected

**(18)** — Line 220: When the letter 'm' is shown in italic font, this reviewer assumes it is a variable, e.g. 25 times m. If it is meant to be a unit (meters), then it should not be in italic font.

    **A —** Done

**(19)** — Line 239: Add spaces between the years in this list.

    **A —** Done

**(20)** — Line 250: "a" can be deleted.

    **A —** Done

**(21)** — Line 268: "hurricane" should be capitalized.

    **A —** Done

**(22)** — Lines 299-300: Can this sentence be rewritten for clarity?

    **A —** .We have slightly rephrased the sentence

**(23) — Line 300: "In fact" can be deleted.**

**A — Done**

**Comments to figures:**

**(24) — Figure 1 caption: In the second sentence, starting with "Where" is awkward - should this instead be a continuation of the first sentence? In the third sentence, the word "represents" should not be plural.**

**A — Done**

**(25) — Figure 3: For panels (a) and (b), why not use 'intensity' or 'maximum wind speeds' as labels for both plots?**

**A — Done**

**(26) — Figure 3 caption: "when if is within" should be corrected.**

**A — Done**

**(27) — Figure 6 caption: "taht" is misspelled.**

**A — Done**

**(28) — Figure 7 caption: "Levels" should not be capitalized.**

**A — Done**